# CONDITIONAL INFORMATION BOTTLENECK APPROACH FOR OUT-OF-DISTRIBUTION SEQUENTIAL RECOMMENDATION

## ABSTRACT

Sequential recommendation (SR) aims to suggest items users are most likely to engage with next based on their past interactions. However, in practice, SR systems often face the out-of-distribution (OOD) problem due to dynamic environmental factors (*e.g.*, seasonal changes), leading to significant performance degradation in the testing phase. Some methods incorporate distributionally robust optimization (DRO) into SR to alleviate OOD, but the sparsity of SR data challenges this. Other approaches use random data augmentations to explore the OOD, potentially distorting important information, as user behavior is personalized rather than random. Additionally, they often overlook users' varying sensitivity to distribution shifts during the exploration, which is crucial for capturing the evolution of user preferences in OOD contexts. In this work, inspired by information bottleneck theory (IB), we propose the Conditional Distribution Information Bottleneck (CDIB), a novel objective that creates diverse OOD distributions while preserving *minimal sufficient information* regarding the origin distribution conditioned on the user. Building on this, we introduce a framework with a learnable, personalized data augmentation method using a mask-then-generate paradigm to craft diverse and reliable OOD distributions optimized with CDIB. Experiments on four real-world datasets show our model consistently outperforms baselines. The code is available at https://anonymous.4open.science/r/CDIB-51C8.

## 1 INTRODUCTION

Nowadays, recommendation systems are important in addressing information overload across various applications, such as e-commerce, online retail platforms, and so on (Cen et al., 2020; Guy et al., 2010). SR is one of the crucial topics focusing on capturing users' dynamic interest to recommend content that aligns with it more accurately (Hidasi et al., 2016; Kang & McAuley, 2018).

Nevertheless, most methods assume that the popularity distribution during training and testing is independent and identically distributed, an unrealistic assumption in most cases (Zheng et al., 2021; Zhang et al., 2023). In SR, popularity distribution can shift due to time-sensitive environmental factors, leading to changes in user preferences (*e.g.*, the World Cup boosting soccer jersey sales or seasonal changes increasing T-shirt sales in summer and sweater sales in winter), which causes performance degradation of the model during the testing phases.

Furthermore, we observe that different users have varying sensitivity to distribution shifts, leading to different impacts from OOD scenarios. As shown in Figure 1, for blockbuster users who engage with trending content, the model can adjust and continue providing relevant recommendations as trends shift (①→③) due to its inherent bias toward popular items (Zhang et al., 2021). However, for niche users who follow mainstream items less, despite the model capturing their preferences during training, it often defaults to providing popular items when environmental factors change, likely due to unfamiliar behavior patterns, misaligning with niche users' true preferences (②→④).

To alleviate the OOD problem in SR, various models have been developed, employing techniques such as reweighting (Wang et al., 2022b), causal inference (Wang et al., 2023b; He et al., 2022), distributionally robust optimization (Yang et al., 2023b; Wen et al., 2022), and contrastive learning (CL) (Liu et al., 2021; Xie et al., 2022; Yang et al., 2023a; Qiu et al., 2022).

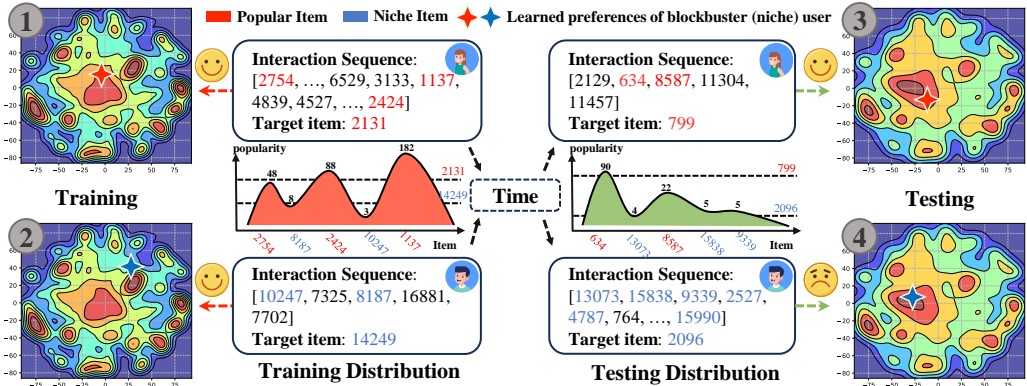

Figure 1: ① and ② show the overall user preference distribution learned by SASRec (Kang & McAuley, 2018) during training, along with learned blockbuster and niche user preferences, respectively, while ③ and ④ display the corresponding distributions during testing. (Blockbuster user: prefers mainstream items; Niche user: prefers mainstream items less.)[1]

However, existing methods have two main issues: **(i)** Methods like DRO in SR (Yang et al., 2023b) optimize the model for the worst-case distribution within a family of distributions around the training data, ensuring robustness to unknown distributions. However, the feasible distribution family for DRO is inherently limited by the sparse nature of recommendation data (Wang et al., 2024). **(ii)** Data augmentation models (*e.g.*, CL methods (Qiu et al., 2022; Xie et al., 2022)) can expand the training distribution but often rely on unguided or hand-crafted augmentations, risking the loss of important interaction data while retaining noisy or irrelevant information for augmentation, misleading user preference modeling. Additionally, they overlook users' varying sensitivity to distribution shifts mentioned above, which is essential for capturing the evolution of user preferences in OOD contexts.

To this end, inspired by IB theory (Gondek & Hofmann, 2003; Tishby & Zaslavsky, 2015; Alemi et al., 2016; Lee et al., 2023; Choi & Lee, 2023), we propose the Conditional Distribution Information Bottleneck (CDIB) that generates diverse distributions while preserving *minimal sufficient information* from the original distribution conditioned on the user. It aims to introduce more interaction patterns influenced by other time-sensitive environmental factors at training, enhancing performance on unknown distributions. Specifically, CDIB diversifies the generated data by minimizing the mutual information between original distribution $\mathcal{D}$ and generated distribution $\tilde{\mathcal{D}}$ given the user features, while retaining personalized critical information by maximizing the mutual information between generated $\tilde{\mathcal{D}}$ and the target items reflecting users' preferences. To control the information flow from $\mathcal{D}$ to $\tilde{\mathcal{D}}$, we propose a mask-then-generate mechanism. First, we introduce a **Learnable Mask** that adaptively identifies stable and sensitive items (*e.g.*, comic book and soccer jersey during the World Cup) in the interaction sequence based on their semantic relatedness. The sensitive items are then perturbed and reconstructed using the proposed **Distribution Generator**, built on the latent diffusion model (Ho et al., 2020; Rombach et al., 2022; Wang et al., 2023a). This process constrains sensitive information transmission while enriching interactions by reconstructing unseen sensitive items. Moreover, we provide theoretical analyses to justify the rationality of the generated $\tilde{\mathcal{D}}$.

To summarize, this work makes the following contributions: **(i)** We introduce CDIB, which guides the generation of diverse and reliable distributions with personalized information, and conduct theoretical analyses to prove its rationality. **(ii)** We propose a mask-then-generate mechanism to constrain sensitive information transmission and enrich sensitive item interactions for better performance in OOD scenarios. **(iii)** Extensive experiments demonstrate the effectiveness and robustness of CDIB.

## 2 PRELIMINARIES

This section begins by outlining the SR scenario, including notations and the problem formulation (Section 2.1). Then, we introduce the IB theory (Section 2.2), a well-established information theory.

---

[1] The concepts and definition of Blockbuster and Niche are derived from (Wen et al., 2022)

## 2.1 SEQUENTIAL RECOMMENDATION PARADIGM

**Notations.** Denote with $\mathcal{U}$ ($u \in \mathcal{U}$) the user set and with $\mathcal{I}$ ($i \in \mathcal{I}$) the item set, where $|\mathcal{U}|$ and $|\mathcal{I}|$ represent the number of users and items, respectively. Each user $u$ is associated with a chronologically ordered interaction sequence $s_u = \left(i_u^1, \cdots, i_u^L\right)$, with $L$ denoting the length of the sequence, and the target is denoted as $\mathbf{Y} = \{i_u^{L+1} \mid u \in \mathcal{U}\}$, where $i_u^{L+1}$ represents the next item to interact with by user $u$. The collection of all interaction sequences is denoted as $\mathcal{S}$ ($s_u \in \mathcal{S}$), which is further partitioned into the training set $\mathcal{S}_{\text{tr}}$ comprising historical interactions, and the test set $\mathcal{S}_{\text{te}}$ containing future interactions. Also, we denote the stable items as $\mathbf{X}_{\text{s}}$ and the (environmental-)sensitive items within the interaction as $\mathbf{X}_{\text{e}}$. Additionally, we define $\mathcal{D}_{\text{tr}}$ and $\mathcal{D}_{\text{te}}$ as the training and testing distributions.

**Problem Formulation.** Formally, the learning of sequential recommendation involves optimizing the model $\xi$ through empirical risk minimization on the training distribution $\mathcal{D}_{\text{tr}}$ (Kang & McAuley, 2018; Sun et al., 2019), which only involves ID features:

$$\xi^* = \arg\min_{\xi} \hat{\mathbb{E}}_{\mathcal{D}_{\text{tr}}} \left[\ell\left(\xi\left(s_u\right), i_u^{L+1}\right)\right] = \arg\min_{\xi} \frac{1}{|\mathcal{S}_{\text{tr}}|} \sum_{s_u \in \mathcal{S}_{\text{tr}}} -\log p\left(i_u^{L+1} \mid \xi\left(s_u\right)\right), \quad (1)$$

where $\xi(s_u)$ represents the hypothesis predicted by $\xi$, and $p\left(i_u^{L+1} \mid \xi\left(s_u\right)\right)$ denotes the probability of $\xi$ recommending $i_u^{L+1}$ to user $u$ based on $s_u$. The optimized $\xi^*$ is then applied to the future task.

## 2.2 INFORMATION BOTTLENECK PRINCIPLE

The information bottleneck principle (Tishby & Zaslavsky, 2015; Alemi et al., 2016) is an approach based on information theory designed to balance the trade-off between compressing a random variable and preserving its *minimum sufficient information* about the target variable. It aims to find a compact representation that retains as much information about the target as possible and discards the target-irrelevant information.

**Definition 2.1 (IB)** *Given input variable* $\mathbf{X}$, *target* $\mathbf{Y}$, *and bottleneck variable* $\mathbf{Z}$, *respectively, the IB aims to compress* $\mathbf{X}$ *to* $\mathbf{Z}$, *while keeping the information relevant for* $\mathbf{Y}$:

$$\min_{\mathbf{Z}} I(\mathbf{X}; \mathbf{Z}) - \beta I(\mathbf{Y}; \mathbf{Z}) \quad (2)$$

*where* $I(\mathbf{U}; \mathbf{V}) = \sum_{u,v} p(u, v) \log \frac{p(u,v)}{p(u)p(v)}$ *is the mutual information between* $\mathbf{U}$ *and* $\mathbf{V}$, *and* $\beta \in \mathbb{R}$ *is a Lagrange multiplier balancing the two mutual information terms.*

# 3 METHODOLOGY

In this section, we first formally propose the DIB principle in Section 3.1, a straightforward application of IB to distribution generation, and then we propose the CDIB in Section 3.2, a novel method that generates diverse and reliable distribution based on conditional mutual information. Then, we introduce the overall model architecture and its optimization strategies with CDIB in Section 3.3.

## 3.1 DISTRIBUTION INFORMATION BOTTLENECK PRINCIPLE

In this section, we introduce the DIB, anchored in the IB. This principle facilitates the generation of new distributions, represented as $\tilde{\mathcal{D}} = (\tilde{\mathbf{X}}_{\text{s}}, \tilde{\mathbf{X}}_{\text{e}})$, derived from the training distribution $\mathcal{D}_{\text{tr}} = (\mathbf{X}_{\text{s}}, \mathbf{X}_{\text{e}})$. With the DIB, while the $\tilde{\mathcal{D}}$ preserves the stable elements within the $\mathcal{D}_{\text{tr}}$, it introduces a spectrum of sensitive elements, thereby enhancing the diversity of the data.

**Definition 3.1 (DIB)** *Given original training distribution* $\mathcal{D}_{\text{tr}}$, *target* $\mathbf{Y}$, *and generated distribution* $\tilde{\mathcal{D}}$, *respectively, we define DIB as follows:*

$$\min_{\tilde{\mathcal{D}}} I(\mathcal{D}_{\text{tr}}; \tilde{\mathcal{D}}) - \beta I(\mathbf{Y}; \tilde{\mathcal{D}}) \quad (3)$$

*where* $\beta \in \mathbb{R}$ *is a Lagrange multiplier balancing the diversity and reliability of the* $\tilde{\mathcal{D}}$.

DIB seeks to foster a variety of distributions $\tilde{\mathcal{D}}$ that diverge from the original distribution by minimizing $I(\mathcal{D}_{\mathrm{tr}}; \tilde{\mathcal{D}})$, while concurrently ensuring the preservation of critical information by maximizing $I(\mathbf{Y}; \tilde{\mathcal{D}})$. It can be demonstrated that this leads to $\tilde{\mathbf{X}}_{\mathrm{s}} \simeq \mathbf{X}_{\mathrm{s}}$ and $\tilde{\mathbf{X}}_{\mathrm{e}} \nsim \mathbf{X}_{\mathrm{e}}$, achieving a balance between stability in stable elements and diversity in sensitive elements (*cf.* Appendix A.1).

### 3.2 Conditional Distribution Information Bottleneck Principle

Although DIB can generate a diverse and promising distribution $\tilde{\mathcal{D}}$, the generation process of the distributions is still not fully reliable due to the lack of constraints. Specifically, there is no certainty that the distributions obtained from the minimization of DIB will represent those encountered in the testing stage. It's because $\tilde{\mathbf{X}}_{\mathrm{e}}$ can be generated in any direction that diverges from $\mathbf{X}_{\mathrm{e}}$ as long as it can minimize the $I(\mathcal{D}_{\mathrm{tr}}, \tilde{\mathcal{D}})$. Furthermore, the straightforward application of DIB fails to account for users' sensitivity to OOD. More concretely, some users are readily influenced by environmental factors like popular trends. In contrast, others display less susceptibility to such influences. Consequently, the generation of $\tilde{\mathbf{X}}_{\mathrm{e}}$ should be personalized, suggesting that the generation of $\tilde{\mathcal{D}}$ should be more controlled. To this end, we introduce the CDIB, aiming to guide the personalized generation of $\tilde{\mathcal{D}}$ and steer it, to a certain extent, towards aligning with the testing distribution:

**Definition 3.2 (CDIB)** *Given original training distribution $\mathcal{D}_{\mathrm{tr}}$, target $\mathbf{Y}$, generated distribution $\tilde{\mathcal{D}}$, and user embeddings $\mathbf{\Gamma}$ respectively. The formulation of CDIB is as follows:*

$$\min_{\tilde{\mathcal{D}}} I(\mathcal{D}_{\mathrm{tr}}; \tilde{\mathcal{D}} \mid \mathbf{\Gamma}) - \beta I(\mathbf{Y}; \tilde{\mathcal{D}} \mid \mathbf{\Gamma}) \tag{4}$$

*where $I(\mathbf{U}; \mathbf{V}|\mathbf{W}) = \sum_w p(w) \sum_{u,v} p(u,v|w) \log \frac{p(u,v|w)}{p(u|w)p(v|w)}$ is the mutual information between $\mathbf{U}$ and $\mathbf{V}$ conditioned on $\mathbf{W}$.*

The first term, $I(\mathcal{D}_{\mathrm{tr}}; \tilde{\mathcal{D}} \mid \mathbf{\Gamma})$, functioning as the conditional generation term, facilitates the personalized separation of $\tilde{\mathcal{D}}$ from $\mathcal{D}_{\mathrm{tr}}$ by minimizing the mutual information between them that takes into account user features. The second term, $I(\mathbf{Y}; \tilde{\mathcal{D}} \mid \mathbf{\Gamma})$, serving as the conditional regularization term, prompts the $\tilde{\mathcal{D}}$ to preserve user-specific target-relevant information from the true labels. By optimizing these two terms, $\tilde{\mathcal{D}}$ includes the *minimum sufficient personalized information* about the target, along with elements that account for user sensitivity. We further demonstrate that introducing additional user features to generate a diverse distribution enhances the model's generalizability.

**Theorem 3.3 (Generalization Bound)** *Let $\hat{\mathbb{E}}_{\mathcal{D}_{\mathrm{tr}}}[\ell(f; \mathbf{Y})]$ be the empirical loss on the training set, $\mathcal{D}$ be the unknown distribution, and $\mathbb{E}_{\mathcal{D}}[\ell(f; \mathbf{Y})]$ be the expected loss on $\mathcal{D}$. Given any finite hypothesis space $\mathcal{F}$ of models, suppose $f \in [M_1, M_2]$, we have that with probability at least $1 - \delta$:*

$$\mathbb{E}_{\mathcal{D}}[\ell(f; \mathbf{Y})] \leq \hat{\mathbb{E}}_{\mathcal{D}_{\mathrm{tr}}}[\ell(f; \mathbf{Y})] + 2\mathcal{R}_n(\mathcal{F}) + (M_2 - M_1)\sqrt{\frac{\log \frac{2}{\delta}}{m}} \tag{5}$$

*where $\mathcal{R}_n(\mathcal{F})$ is the rademacher complexity of $\mathcal{F}$, reflecting its capacity to model random noise within a dataset, inherently linked to the dataset's properties, and m is the amount of the user features.*

The proof is presented in the Appendix A.2. Theorem 3.3 shows that an increased value of $m$ results in a tighter bound for $\mathbb{E}_{\mathcal{D}}[\ell(f; \mathbf{Y})]$, which is upper bounded by $\hat{\mathbb{E}}_{\mathcal{D}_{\mathrm{tr}}}[\ell(f; \mathbf{Y})]$, thereby enhancing the model's generalizability in unknown distribution.

### 3.3 Model Architecture and Optimization

In this section, we formally introduce the generation process for the distribution $\tilde{\mathcal{D}}$ (Section 3.3.1). Following this, we detail the optimization with the CDIB (Section 3.3.2).

#### 3.3.1 Generation of Distribution $\tilde{\mathcal{D}}$

Current graph IB-based approaches inject Gaussian noise into or mask insignificant nodes to control the information flow from the original $\mathcal{D}_{\mathrm{tr}}$ to the generated $\tilde{\mathcal{D}}$ (Wei et al., 2022; Lee et al., 2023).

Figure 2: The overall framework of CDIB: The learnable mask first masks the stable items, followed by the distribution generator augmenting the sensitive items. Both original and augmented samples are then fed into the recommender to obtain $\mathcal{D}$ and $\tilde{\mathcal{D}}$, which are optimized using CDIB later.

However, these methods rely solely on a node's embedding to determine significance, hard to perceive the current interaction environment and distinguish between stable and sensitive items. Furthermore, adding Gaussian noise often disrupts semantic integrity, and struggle to simulate OOD samples well. Thus, we propose a mask-then-generate mechanism consisting of: **(i) Learnable Mask**-adaptively discover sensitive items; **(ii) Distribution Generator**-perturb and reconstruct sensitive items.

**Embedding Layer.** Users and items are embedded into a $d$-dimensional latent space. For each user $u$ with interaction sequence $s_u$, we obtain a user embedding $\mathbf{e}_u \in \mathbb{R}^d$ and an item embedding matrix $\mathbf{E}_u = \left(\mathbf{e}_u^1, \ldots, \mathbf{e}_u^L\right) \in \mathbb{R}^{L \times d}$, where $\mathbf{e}_u^l$ represents the $l$-th item interacted by $u$. The set of all user embeddings is denoted as $\mathbf{\Gamma} = \{\mathbf{e}_u \mid u \in \mathcal{U}\} \in \mathbb{R}^{|\mathcal{U}| \times d}$. To model temporal information, we initialize a learnable position embedding matrix $\mathbf{P} = (\mathbf{p}_1, \ldots, \mathbf{p}_L) \in \mathbb{R}^{L \times d}$, commonly used in sequence modeling (Devlin et al., 2018; Sun et al., 2019). The hidden representation for a sequence is then computed as $\mathbf{H}_u = \mathbf{E}_u + \mathbf{P}$, where $\mathbf{h}_u^l \in \mathbf{H}_u$. To model the users' sensitivity, we minimize KL-divergence to align the user feature distribution with the interacted items' popularity distribution:

$$\mathcal{L}_{\mathrm{con}} = \mathbb{E}_{\mathcal{D}_{\mathrm{tr}}}\left[D_{KL}(\mathcal{N}\left(p\left(\mathbf{\Gamma}\right), \mathbf{I}\right) \parallel \mathcal{N}((O_{\mathbf{Y}} / \textstyle\sum O_{\mathbf{Y}}), \mathbf{I}))\right], \qquad (6)$$

where $O_{\mathbf{Y}}$ represents the number of times each target has been observed, and $p(\cdot)$ is the sensitivity estimator, which is implemented using an MLP.

**Learnable Mask.** Inspired by the InfoMax principle (Ye et al., 2023), we measure semantic relatedness by assessing representation consistency within the interaction sequence. Sensitive items, with uncertain behavioral patterns, often show lower semantic relatedness due to embedding differences from other items. Thus, items with low semantic relatedness are classified as sensitive. We compute this relatedness using an MLP parameterized by $\theta_1$, summarized as follows:

$$\mathbf{M}_u = \left[\mathbf{M}_u^1, \cdots, \mathbf{M}_u^l, \cdots, \mathbf{M}_u^L\right], \text{ where } \mathbf{M}_u^l = \sigma\left(\mathrm{MLP}_{\theta_1}\left(\mathbf{h}_u^l \parallel \phi\left(\mathbf{H}_u\right) \parallel \mathbf{e}_u\right)\right), \qquad (7)$$

where $\phi(\cdot)$ represents the interaction aggregation function, we have chosen to implement the mean aggregation. $\sigma(\cdot)$ denotes the sigmoid function. To avoid $\mathbf{M}_u$ converging to trivial solutions $\mathbf{0}$, we introduce a self-supervised regularization loss: $\mathcal{L}_{\mathrm{mask}} = -\sum_{u \in \mathcal{U}} \sum_{l=1}^{L} \mathbf{M}_u^l$. Also, the optimized $\mathcal{L}_{\mathrm{mask}}$ dynamically adjusts semantic relatedness based on the impact of masking on downstream tasks, enabling adaptive masking in evolving SR scenarios.

**Distribution Generator.** After distinguishing stable and sensitive items, we perturb sensitive items to compress stable information into $\tilde{\mathcal{D}}$. Instead of simply injecting noise, we employ the diffusion model paradigm (Rombach et al., 2022) to progressively add noise to sensitive item representations until they approximate a normal distribution, followed by denoising to reconstruct unseen sensitive items embeddings. These high-probability yet unobserved interactions mitigate dataset sparsity, improving prediction accuracy. Concretely, we first mask the stable items by $\mathbf{H}_u^0 = \mathbf{H}_u \odot \mathbf{M}_u$, where $\odot$ is the broadcasted element-wise product, and then we incrementally introduced Gaussian noise into it, creating a sequence $\mathbf{H}_u^{1:T}$ through $T$ steps in a Markov chain, which can be formulated as follows:

$$q\left(\mathbf{H}_u^t \mid \mathbf{H}_u^{t-1}\right) = \mathcal{N}\left(\mathbf{H}_u^t; \sqrt{1 - \beta_t}\mathbf{H}_u^{t-1}, \beta_t \mathbf{I}\right), \qquad (8)$$

where $\mathcal{N}$ indicates the Gaussian distribution and $\beta_t \in (0, 1)$ specifies the scale of noise introduced at each step $t$. Through the reparameterization trick and principle that the sum of two independent

Gaussian noises is also Gaussian, $\mathbf{H}_u^t$ can be directly derived from $\mathbf{H}_u^0$ as $\mathbf{H}_u^t = \sqrt{\bar{\alpha}_t}\mathbf{H}_u^0 + \sqrt{1-\bar{\alpha}_t}\epsilon_t$, with $\epsilon_t \sim \mathcal{N}(0, \mathbf{I})$ as the added noise, and $\bar{\alpha}_t = \prod_{t'=1}^T (1-\beta_{t'})$. After that, CDIB iteratively remove the noise from $\mathbf{H}_u^t$ to reconstruct $\mathbf{H}_u^{t-1}$ and ultimately recover the original sample $\mathbf{H}_u^0$:

$$p_{\theta_2}\left(\mathbf{H}_u^{0:T}\right) = p\left(\mathbf{H}_u^T\right)\prod_{t=1}^T p_{\theta_2}\left(\mathbf{H}_u^{t-1} \mid \mathbf{H}_u^t\right), \tag{9}$$

where $p\left(\mathbf{H}_u^T\right) \sim \mathcal{N}(0, \mathbf{I})$ and $\prod_{t=1}^T p_{\theta_2}\left(\mathbf{H}_u^{t-1} \mid \mathbf{H}_u^t\right)$ denotes the process of sequentially deducing $\mathbf{H}_u^{t-1}$ by reversing the estimated Gaussian noise from $\mathbf{H}_u^t$ via a lightweight MLP network parameterized by $\theta_2$. The learning objective is thereby distilled to:

$$\mathcal{L}_{\text{diff}} = \sum_{t=2}^T \mathbb{E}_{t,\epsilon}\left[\|\epsilon_t - \epsilon_{\theta_2}\left(\mathbf{H}_u^t, t\right)\|_2^2\right], \tag{10}$$

where $\epsilon_t$ represents the noise have been added to $\mathbf{H}_u^{t-1}$ in the forward process. Then, CDIB generate unseen sensitive items by firstly corrupting $\mathbf{H}_u^0$ via Equation (36), and executing reverse denoising on corrupted representation via Equation (37) to obtain the rich sensitive elements denoted as $\tilde{\mathbf{H}}_u^0$, then we obtain the generated sequence embedding as $\tilde{\mathbf{H}}_u = \tilde{\mathbf{H}}_u^0 + \mathbf{H}_u \odot (1 - \mathbf{M}_u)$. Finally, the total loss for generating $\tilde{\mathbf{H}}_u$ is given by: $\mathcal{L}_{\text{gd}} = \mathcal{L}_{\text{con}} + \mathcal{L}_{\text{mask}} + \mathcal{L}_{\text{diff}}$.

**Transformer Recommender.** Using the Learnable Mask and Distribution Generator, we generate the sequence embedding $\tilde{\mathbf{H}}_u$. Next, we apply a multi-head self-attention mechanism to refine both the original embedding $\mathbf{H}_u$ and the generated embedding $\tilde{\mathbf{H}}_u$ (Devlin et al., 2018; Vaswani et al., 2023):

$$\dot{\mathbf{H}}_u = \varphi\left(\Big\|_{h=1}^H \bar{\mathbf{H}}_u^h\right); \quad \bar{\mathbf{H}}_u^h = \text{softmax}\left(\mathbf{H}_u\mathbf{W}_Q^h(\mathbf{H}_u\mathbf{W}_K^h)^T/\sqrt{d/h}\right)\mathbf{H}_u\mathbf{W}_V^h, \tag{11}$$

where $\varphi(\mathbf{x}) = \text{GELU}(\mathbf{W}\mathbf{x} + \mathbf{b})$, and $\dot{\mathbf{H}}_u \in \mathbb{R}^{L \times d}$ is the refined item embedding, derived by concatenating $\mathbf{H}_u^h \in \mathbb{R}^{L \times d/H}$ and applying $\varphi(\cdot)$. The refined embedding of $\tilde{\mathbf{H}}_u$ is denoted as $\ddot{\mathbf{H}}_u$. For both $\dot{\mathbf{H}}_u$ and $\ddot{\mathbf{H}}_u$, the last position vector represents the entire interaction sequence (Kang & McAuley, 2018), denoted as $\dot{\mathbf{h}}_u$ and $\ddot{\mathbf{h}}_u$, respectively, with the same label (*i.e.*, $i_u^{L+1}$). The distributions of $\dot{\mathbf{h}}_u$ and $\ddot{\mathbf{h}}_u$ are $\mathcal{D}_{\text{tr}}$ and $\tilde{\mathcal{D}}$, respectively.

### 3.3.2 MODEL OPTIMIZATION WITH CDIB

**Maximizing $I(\mathbf{Y}; \tilde{\mathcal{D}}|\Gamma)$.** Directly maximizing the conditional regularization term proves challenging. Hence, according to (Choi & Lee, 2023), we instead derive and maximize the lower bound of $I(\mathbf{Y}; \tilde{\mathcal{D}}, \Gamma)$ via variational decomposition (*cf.* Appendix A.3), outlined as follows:

**Proposition 3.4 (Lower bound of $I(\mathbf{Y}; \tilde{\mathcal{D}}, \Gamma)$)** *Given label $\mathbf{Y}$, distribution $\tilde{\mathcal{D}}$, and user features $\Gamma$:*

$$I\left(\mathbf{Y}; \tilde{\mathcal{D}}, \Gamma\right) \geq \mathbb{E}_{\mathbf{Y}, \tilde{\mathcal{D}}, \Gamma}\left[\log p_{\theta_3}\left(\mathbf{Y} \mid \tilde{\mathcal{D}}, \Gamma\right)\right] \tag{12}$$

*where $\log p_{\theta_3}(\mathbf{Y} \mid \tilde{\mathcal{D}}, \Gamma)$ is the variational approximation of $\log p(\mathbf{Y} \mid \tilde{\mathcal{D}}, \Gamma)$.*

Given that the generation process of $\tilde{\mathcal{D}}$ incorporates user embeddings $\Gamma$ as defined in Equation (7), we have $\log p_{\theta_3}(\mathbf{Y}|\tilde{\mathcal{D}}, \Gamma) = \log p_{\theta_3}(\mathbf{Y}|\tilde{\mathcal{D}})$. Here, $\log p_{\theta_3}(\mathbf{Y}|\tilde{\mathcal{D}})$ represents a recommendation task, where the input is the generated distribution $\ddot{\mathbf{h}}_u \in \tilde{\mathcal{D}}$ and the output is the next interacted item $i_u^{L+1} \in \mathbf{Y}$. We optimize it using the transformer recommender $f_{\theta_3}$ with the objective defined in Equation (1), employing cross-entropy loss over the full set of items, and the loss is denoted as $\mathcal{L}_{\text{reg}}$.

**Minimizing $I(\mathcal{D}_{\text{tr}}; \tilde{\mathcal{D}}|\Gamma)$.** To minimize the conditional generation term, we first employ the chain rule for mutual information[2], applying it as follows: $I(\mathcal{D}_{\text{tr}}; \tilde{\mathcal{D}}|\Gamma) = I(\mathcal{D}_{\text{tr}}; \tilde{\mathcal{D}}, \Gamma) - I(\mathcal{D}_{\text{tr}}; \Gamma)$. Notice that $I(\mathcal{D}_{\text{tr}}; \tilde{\mathcal{D}}, \Gamma) = I(\mathcal{D}_{\text{tr}}; \tilde{\mathcal{D}})$ (*cf.* Appendix A.4). Intuitively, minimizing the first term personalized drives $\tilde{\mathcal{D}}$ away from $\mathcal{D}_{\text{tr}}$, thereby fostering a diverse distribution exploration. Maximizing the second term seeks to capture the personalized information from $\mathcal{D}_{\text{tr}}$ into $\Gamma$. Inspired by (Wei et al., 2022), we adopt negative InfoNCE to estimate the mutual information (Gutmann & Hyvärinen, 2010) and

---

[2]Given the random variables $\mathbf{X}$, $\mathbf{V}$, and $\mathbf{Z}$, then the chain rule gives $I(\mathbf{X}; \mathbf{V}|\mathbf{Z}) = I(\mathbf{X}, \mathbf{Z}; \mathbf{V}) - I(\mathbf{Z}; \mathbf{V})$.

contrastive learning to minimize the $I(\mathcal{D}_{\mathrm{tr}}; \tilde{\mathcal{D}}|\mathbf{\Gamma})$. Specifically, for the $I(\mathcal{D}_{\mathrm{tr}}; \tilde{\mathcal{D}})$, we treat the original sequence embedding $\dot{\mathbf{h}}_u \in \mathcal{D}_{\mathrm{tr}}$ and the corresponding augmented sequence embedding $\ddot{\mathbf{h}}_u \in \tilde{\mathcal{D}}$ as positive pairs, with in-batch instances serving as negative samples. For $I(\mathcal{D}_{\mathrm{tr}}; \mathbf{\Gamma})$, the original sequence embedding $\dot{\mathbf{h}}_u$ and the corresponding user embedding $\mathbf{e}_u$ are considered positive pairs, again with in-batch instances as negative samples. We define the contrastive loss as follows:

$$\mathcal{L}_{\mathrm{gen}} = \frac{1}{|\mathcal{U}|} \sum_{u \in \mathcal{U}} \left( \log \frac{e^{\phi(\dot{\mathbf{h}}_u, \ddot{\mathbf{h}}_u)/\tau}}{\sum_{u' \in \mathcal{U}} e^{\phi(\dot{\mathbf{h}}_u, \ddot{\mathbf{h}}_{u'})/\tau)}} - \log \frac{e^{\phi(\dot{\mathbf{h}}_u, \mathbf{e}_u)/\tau}}{\sum_{u' \in \mathcal{U}} e^{\phi(\dot{\mathbf{h}}_u, \mathbf{e}_{u'})/\tau)}} \right), \tag{13}$$

where $\phi(\cdot)$ denotes the similarity function and $\tau$ denotes the tunable hyper-parameter to adjust the scale for softmax.

**Overall Objective.** Finally, we train the model using the specified final objective as follows:

$$\mathcal{L}_{\mathrm{total}} = \mathcal{L}_{\mathrm{pred}} + \alpha_1 \mathcal{L}_{\mathrm{gd}} + \alpha_2 \left( \beta \mathcal{L}_{\mathrm{reg}} + \mathcal{L}_{\mathrm{gen}} \right), \tag{14}$$

where $\mathcal{L}_{pred}$ is the primary recommendation loss, calculated by the $f_{\theta_3}$ (which is also employed to optimize the conditional regularization term), where the input is the training data ($\dot{\mathbf{h}}_u \in \mathcal{D}_{\mathrm{tr}}$), and the output is the next item interacted with. The $\alpha_1, \alpha_2$ represent tunable hyperparameters that balance the significance of auxiliary losses. Ultimately, the trained $f_{\theta_3}$ serves as the $\xi^*$ in the testing stage.

# 4 EXPERIMENT

**Datasets.** Our experiments are conducted on four real-world datasets, *i.e.*, `ML100K`, `Retail`, `Beauty`, and `Sports`. For each dataset, we chronologically select 80% of the historical interactions of each user as the training set, 10% of those as the validation set, and the remaining 10% as the test set. The detailed information is in Appendix E.1.

**Baselines.** We compare CDIB with nine methods from diverse research lines, covering (i) Naive Sequential Recommendation Methods: **GRU4Rec** (Hidasi et al., 2016), **Caser** (Tang & Wang, 2018), and **SASRec** (Kang & McAuley, 2018). (ii) Reweighting Methods: **IPS** (Schnabel et al., 2010). (iii) DRO Methods: **S-DRO** (Wen et al., 2022) and **DROS** (Yang et al., 2023b). (iv) Diffusion-based Augmentation Methods: **DiffuASR-CG** and **DiffuASR-CF** (Liu et al., 2023). (v) Contrastive Learning (CL) Methods: **CL4SRec** (Xie et al., 2022), **DuoRec** (Qiu et al., 2022), and **DCRec** (Yang et al., 2023a). The details are in Appendix E.3.

## 4.1 PERFORMANCE COMPARISON

**Overall Performance Comparison.** We assess the methods using the all-ranking protocol (He et al., 2020), focusing on HR@10 and NDCG@10 metrics. The results are shown in Table 1, and we have several observations: **(i)** The DRO and CL methods outperform naive sequential recommendation models, demonstrating their effectiveness. Specifically, compared with SASRec, DROS shows improvements on the `ML100K` and `Sports`, while DuoRec progresses on the `Retail` and `Beauty`. However, IPS and S-DRO only achieve marginal improvements or perform worse, suggesting their limitations when dealing with sparse data. **(ii)** The efficacy of CL methods appeared to be hindered on the `Sports`, whose average interaction sequence length is the shortest. This indicates a sensitivity to hand-crafted data augmentation, which may limit the success of CL methods. **(iii)** Our model consistently outperforms the baseline models across all datasets, showing the effectiveness of the learnable data augmentation method and the optimization strategy with CDIB, which can create diverse and promising distributions and capture more robust information.

**Robustness to Distribution Shift.** To further evaluate the robustness of our model to distribution shifts, we conduct experiments and compare its performance to that of representative models across different time gaps on `ML100K` and `Retail` datasets. The results are shown in Figure 3, where T1 denotes the training stage, and T2 through T7 represents the testing stages, each with an increasing time gap. As the gap size increases, the overall accuracy of the baseline models generally shows a downward trend, highlighting the severe negative impact of temporal distribution shifts. For example, SASRec's performance drop $72.48\%$ from T1 to T7 under the `Retail` datasets, whereas CDIB remains more stable with the drop rate of $67.97\%$. We attribute it to the fact that generated distribution allows the model to recognize and adapt to these OOD situations at the training stage.

Table 1: Overall performance. The best results and second-best are in **bold** and underline. All the numbers are percentage values with "%" omitted (mean±std). ♣ is the model's variants in the ablation study. The experiments are conducted 5 times.

| Method | MovieLens-100K | | Retail | | Amazon-Beauty | | Amazon-Sports | |
|---|---|---|---|---|---|---|---|---|
| | HitRate ↑ | NDCG ↑ | HitRate ↑ | NDCG ↑ | HitRate ↑ | NDCG ↑ | HitRate ↑ | NDCG ↑ |
| GRU4Rec | $10.26_{\pm0.22}$ | $4.90_{\pm0.09}$ | $12.03_{\pm0.29}$ | $5.90_{\pm0.10}$ | $6.49_{\pm0.27}$ | $3.44_{\pm0.16}$ | $3.47_{\pm0.15}$ | $1.80_{\pm0.10}$ |
| Caser | $6.22_{\pm0.39}$ | $2.90_{\pm0.21}$ | $7.28_{\pm0.26}$ | $3.16_{\pm0.18}$ | $3.74_{\pm0.13}$ | $1.83_{\pm0.07}$ | $2.02_{\pm0.11}$ | $1.00_{\pm0.07}$ |
| SASRec | $10.96_{\pm0.12}$ | $4.84_{\pm0.05}$ | $19.78_{\pm0.14}$ | $8.67_{\pm0.07}$ | $8.64_{\pm0.13}$ | $4.29_{\pm0.06}$ | $4.76_{\pm0.05}$ | $2.22_{\pm0.02}$ |
| IPS | $10.97_{\pm0.10}$ | $4.85_{\pm0.03}$ | $19.65_{\pm0.16}$ | $8.60_{\pm0.08}$ | $8.71_{\pm0.08}$ | $4.31_{\pm0.03}$ | $4.74_{\pm0.07}$ | $2.21_{\pm0.03}$ |
| S-DRO | $10.90_{\pm0.13}$ | $4.82_{\pm0.05}$ | $19.70_{\pm0.20}$ | $8.64_{\pm0.09}$ | $8.63_{\pm0.14}$ | $4.27_{\pm0.06}$ | $4.74_{\pm0.07}$ | $2.22_{\pm0.02}$ |
| DROS | $11.30_{\pm0.11}$ | $5.23_{\pm0.06}$ | $18.79_{\pm0.16}$ | $8.65_{\pm0.07}$ | $8.33_{\pm0.13}$ | $4.14_{\pm0.10}$ | $4.81_{\pm0.07}$ | $2.32_{\pm0.06}$ |
| DiffuASR-CG | $11.18_{\pm0.22}$ | $5.13_{\pm0.07}$ | $20.31_{\pm0.21}$ | $8.84_{\pm0.09}$ | $8.33_{\pm0.12}$ | $4.13_{\pm0.10}$ | $4.70_{\pm0.09}$ | $2.19_{\pm0.04}$ |
| DiffuASR-CF | $11.24_{\pm0.21}$ | $5.19_{\pm0.06}$ | $20.51_{\pm0.23}$ | $8.97_{\pm0.12}$ | $8.46_{\pm0.12}$ | $4.26_{\pm0.08}$ | $4.79_{\pm0.14}$ | $2.23_{\pm0.05}$ |
| CL4SRec | $11.07_{\pm0.35}$ | $5.16_{\pm0.07}$ | $19.72_{\pm0.26}$ | $8.67_{\pm0.10}$ | $\underline{8.80}_{\pm0.06}$ | $\underline{4.39}_{\pm0.05}$ | $4.77_{\pm0.14}$ | $2.26_{\pm0.04}$ |
| DuoRec | $11.21_{\pm0.17}$ | $5.17_{\pm0.06}$ | $\underline{20.63}_{\pm0.11}$ | $\underline{9.10}_{\pm0.06}$ | $8.74_{\pm0.41}$ | $\underline{4.41}_{\pm0.19}$ | $4.49_{\pm0.10}$ | $2.21_{\pm0.04}$ |
| DCRec | $10.63_{\pm0.50}$ | $4.50_{\pm0.34}$ | $20.22_{\pm0.31}$ | $8.82_{\pm0.11}$ | $7.99_{\pm0.38}$ | $3.90_{\pm0.24}$ | $4.08_{\pm0.39}$ | $1.97_{\pm0.16}$ |
| w/o LM♣ | $10.32_{\pm0.27}$ | $4.90_{\pm0.12}$ | $20.01_{\pm0.23}$ | $9.03_{\pm0.13}$ | $8.56_{\pm0.31}$ | $4.26_{\pm0.13}$ | $4.54_{\pm0.17}$ | $2.27_{\pm0.05}$ |
| w/o DG♣ | $11.59_{\pm0.16}$ | $5.53_{\pm0.12}$ | $20.82_{\pm0.16}$ | $9.37_{\pm0.04}$ | $9.06_{\pm0.15}$ | $\mathbf{4.60}_{\pm0.06}$ | $4.92_{\pm0.21}$ | $\mathbf{2.40}_{\pm0.08}$ |
| w/o IB♣ | $10.98_{\pm0.19}$ | $4.88_{\pm0.12}$ | $19.72_{\pm0.32}$ | $8.62_{\pm0.14}$ | $8.68_{\pm0.13}$ | $4.32_{\pm0.04}$ | $4.63_{\pm0.08}$ | $2.17_{\pm0.03}$ |
| CDIB (Ours) | $\mathbf{11.89}_{\pm0.16}$ | $\mathbf{5.67}_{\pm0.09}$ | $\mathbf{21.12}_{\pm0.14}$ | $\mathbf{9.41}_{\pm0.10}$ | $\mathbf{9.17}_{\pm0.04}$ | $4.56_{\pm0.04}$ | $\mathbf{4.95}_{\pm0.11}$ | $2.38_{\pm0.07}$ |

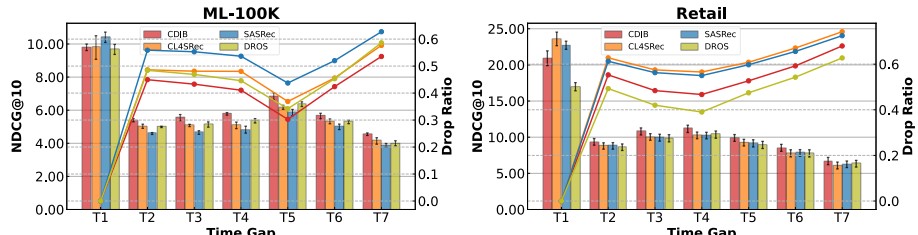

Figure 3: Model performance with respect to time gap on `ML100K` (left) and `Retail` (right), with the time gap relative to T1 increasing from T1 to T7.

**Performance on Different User Group.** We investigated the model's effectiveness across different user groups on `ML100K` and `Retail` datasets as shown in Figure 4. We classify users as niche (U1), diverse (U2→U4), or blockbuster (U5) users based on the proportion of popular items they interact with, and the user conformity increases progressively from U1 to U5; a "blockbuster" user is inclined to follow items that are currently trending, whereas a "niche" user seek to prefer long-tail items. Beyond these two extremes, a "diverse" user group has a broad taste in both popular and long-tail items. The experimental results reveal that the model's effectiveness declines as user conformity decreases, indicating the model's vulnerability to the influence of item popularity while neglecting individual user attributes. Throughout these tests, our model is basically superior to the baseline models. We attribute this superior performance to our optimization strategy, which utilizes user attributes to guide model optimization. This approach allows the model to better capture users' personalized interest and recommend more relevant content.

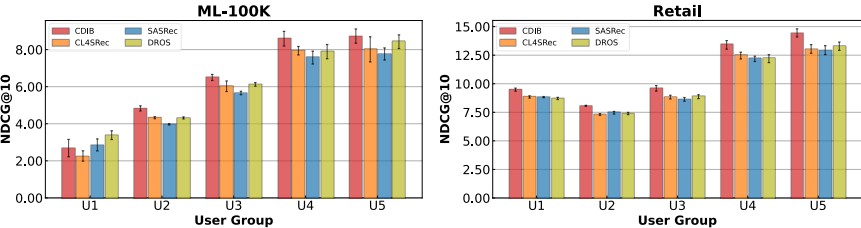

Figure 4: Model performance with respect to user groups on `ML100K` (left) and `Retail` (right), with user conformity increasing progressively from U1 to U5.

## 4.2 Sensitivity Analysis on $\beta$

In this section, we analyze the model's sensitivity to $\beta$, which controls the trade-off between out-of-distribution exploration and prediction accuracy. The results are shown in Figure 5. Our observations are as follows: **(i)** The model fails to converge when $\beta \leq 1e1$. This issue arises because such low values of $\beta$ encourage the model to aggressively generate distributions beyond the training distribution's scope without preserving stable factors, thereby introducing harmful noise. **(ii)** As $\beta$ increases from $1e1$ to $1e3$, performance improves. However, when $\beta$

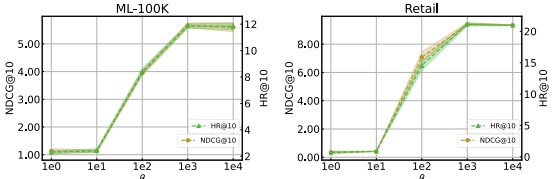

Figure 5: Sensitivity Analysis on $\beta$ under `ML100K` and `Retail` datasets.

reaches $1e4$, performance declines, possibly due to the model's excessive focus on prediction at the expense of sufficient out-of-distribution exploration. Thus, a tailored $\beta$ is needed to balance the two.

## 4.3 Ablation Study

In this section, we explore the design rationale of sub-modules within our CDIB framework. We remove key modules to implement three variants of CDIB: **(i)** "w/o LM": CDIB without the learnable mask, setting $\mathbf{M}_u = \mathbf{1}$. **(ii)** "w/o IB": CDIB without optimization with CDIB, using contrastive learning with InfoNCE loss instead. **(iii)** "w/o DG": CDIB without latent diffusion. Gaussian noise is added to $\mathbf{H}_u$ to get $\tilde{\mathbf{H}}_u$ (*i.e.*, $\tilde{\mathbf{H}}_u = \mathcal{N}(0, \mathbf{I}) \odot \mathbf{M}_u + \mathbf{H}_u \odot (1 - \mathbf{M}_u)$). From the results (Table 1), we observe that: **(i)** Removing the learnable mask significantly degrades performance, underscoring its essential role in identifying elements to be disturbed during data augmentation. Without this component, the model may fail to capture genuine user interest reflected in the interaction sequence, potentially leading to misguided model optimization. **(ii)** Removing latent diffusion for generating distribution shows a performance decline. However, on the `Beauty` and `Sports` datasets, where the average sequence length is the shortest, w/o DG performs better. **(iii)** The gap in performance between CDIB and w/o IB highlights its effectiveness in guiding the distribution generation process and boosting the model's generalization. The performance of w/o IB closely matches that of SASRec, which can be attributed to maximization of $I(\mathcal{D}_{\text{tr}}; \tilde{\mathcal{D}})$ in the standard contrastive learning with InfoNCE loss. Specifically, $\mathbf{M}_u$ may converge on trivial solutions $\mathbf{0}$ to fulfill the CL task, leading to $\mathcal{D}_{\text{tr}} = \tilde{\mathcal{D}}$. This indicates no OOD exploration, the same as SASRec.

## 4.4 Visualization of Blockbuster and Niche Users' Preference

We visualized the interest distributions of blockbuster and niche users learned by CDIB on the `ML100K` and `Retail` datasets during the testing stage. For both SASRec and CDIB, the preference distributions of blockbuster users exhibit significant clustering, likely around popular items (hotspots). However, for niche users, CDIB, compared to SASRec, shows a more uniform preference distribution and is less influenced by popular items, indicating that CDIB effectively models niche users without being affected by new trends, showing the rationality of our model design.

## 5 Related Work

**Sequential Recommendation** is designed to predict the next item a user is likely to prefer based on their interaction history. Traditional methods have leveraged Markov chains to capture first-order item-to-item correlations through transition matrices (Rendle et al., 2010; He & McAuley, 2016). With the development of deep learning, which excels at modeling complex sequential patterns, various deep recommendation models have been developed. For instance, GRU4Rec (Hidasi et al., 2016) employs Gated Recurrent Unit (GRU) units to model the temporal dynamics of interaction sequences. SASRec (Kang & McAuley, 2018) and BERT4Rec (Sun et al., 2019) enhance computational efficiency in lengthy sequences by incorporating self-attention mechanisms. More recently, inspired by selective state space models (Gu & Dao, 2024), Mamba4Rec (Liu et al., 2024a) has been introduced, utilizing the mamba framework to recommend items efficiently. Despite their capabilities, these models often suffer performance declines when OOD occurs. To address this, CDIB introduces a user

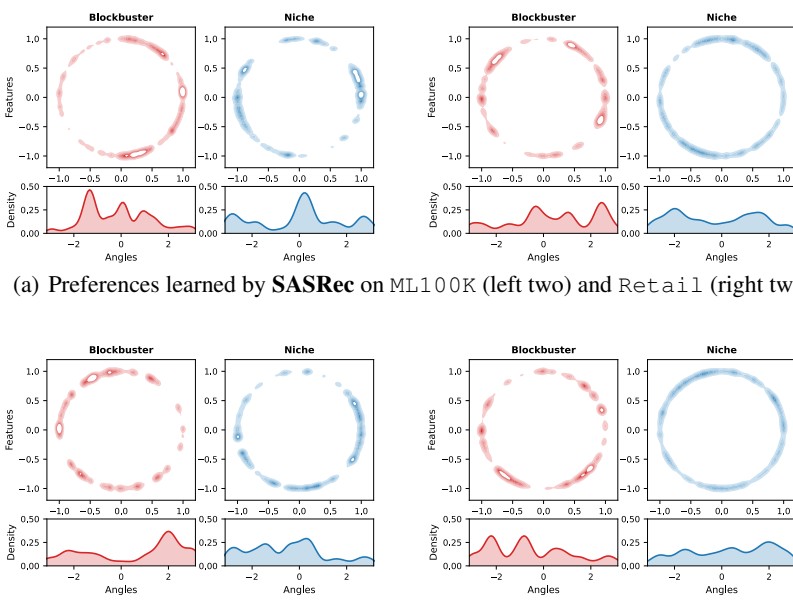

(a) Preferences learned by **SASRec** on `ML100K` (left two) and `Retail` (right two).

(b) Preferences learned by **CDIB** on `ML100K` (left two) and `Retail` (right two)

Figure 6: We visualize preference distributions using Gaussian kernel density estimation (KDE) in $\mathbb{R}^2$ and von Mises-Fisher (vMF) KDE for angular data (*i.e.*, $\arctan 2(y, x)$ for each point (x, y)).

feature-guided generation approach that proactively explores OOD scenarios during the training phase, enhancing the model's generalization capabilities.

**Distributionally Robust Sequential Recommendation** has recently attracted significant research interest, which aims to train a model that performs well not only at the training stage but also at the testing stage. Methods DRO (Schnabel et al., 2010; Bottou et al., 2013; Wang et al., 2022b; Yang et al., 2023b; Wen et al., 2022) optimize the model for the worst-case distribution to improve the robustness. For example, DROS (Yang et al., 2023b) unifies the DRO and sequential recommendation paradigms to enhance model robustness against distribution shifts. Causal inference methods capture real causal relationships but assume the causal graph is static (Wang et al., 2023b; He et al., 2022; Yang et al., 2020; Wang et al., 2022a), these methods face challenges with sparse data. While contrastive learning approaches seek to enrich the training data distribution through data augmentation (Liu et al., 2021; Xie et al., 2022; Yang et al., 2023a; Qiu et al., 2022; Zhao et al., 2023), but hardly rely on the hand-crafted data augmentation strategies. To fill the gap, we introduce the CDIB principle, using the user features to guide the exploration of the other distribution.

**Information Bottleneck with Conditional Information** is also widely utilized. The CIB (Gondek & Hofmann, 2003) theory has been applied in CGIB (Lee et al., 2023) to identify key structures in molecules that predict interaction behaviors between graph pairs, focusing on important subgraphs. Additionally, TimeCIB (Choi & Lee, 2023) extends the CIB to impute time series data, preserving vital temporal information. To the best of our knowledge, CDIB marks the first use of CIB to guide the distribution generation process. The detailed introduction of related works is in Appendix E.4.

## 6 CONCLUSION

In this work, to address the limitations of existing methods that struggle with sparse data or depend on hand-crafted augmentations, we introduce CDIB, an innovative principle that guides the generation of diverse and reliable distributions based on user features. Theoretical analyses demonstrate the rationality of this approach. Building on CDIB, we propose a framework that employs a learnable method to generate distributions for OOD exploration, guided by a conditional generation term and a conditional regularization term. Extensive experiments on four public datasets confirm the effectiveness and robustness of our model.

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

## A  PROOFS

### A.1  PROOF OF RATIONALITY OF DIB

*Proof.* Before the proof, we propose the following two assumptions:

(a) $\mathbf{X}_s$ and $\mathbf{Y}_s$ follow the same distribution.

(b) Since $\tilde{\mathcal{D}}$ is generated by latent diffusion model without introducing any information from $\mathbf{Y}$, $\tilde{\mathbf{X}}_e$ and $\mathbf{Y}_e$ are mutually independent.

Notice that $\mathbf{Y}_s$ and $\mathbf{Y}_e$ refer to the sensitive and stable features of the target item, respectively. For instance, considering a pair of shorts, $\mathbf{Y}_s$ would denote its stable features, such as the brand, while $\mathbf{Y}_e$ would indicate sensitive features like their status as a seasonal trend in summer.

We then obtain the fact that all stable factors and external factors are orthogonal. If this does not stand, $\exists\, x_s \in \mathbf{X}_s, x_e \in \mathbf{X}_e\ s.t.\ corr(x_s, x_e) \neq 0$. When all external features other than $x_e$ remain unchanged while $x_e$ changes, $x_s$ changes correspondingly. This contradicts with $\mathbf{X}_s$ is stable.

We are now ready to prove the proposition. With this fact, we can derive the following:

$$
\begin{aligned}
&I(\mathcal{D}_{tr}; \tilde{\mathcal{D}}) - \beta I(\mathbf{Y}; \tilde{\mathcal{D}}) \\
&= I(\mathbf{X}_s, \mathbf{X}_e; \tilde{\mathbf{X}}_s, \tilde{\mathbf{X}}_e) - \beta I(\mathbf{Y}_s, \mathbf{Y}_e; \tilde{\mathbf{X}}_s, \tilde{\mathbf{X}}_e) \\
&= I(\mathbf{X}_s; \tilde{\mathbf{X}}_s, \tilde{\mathbf{X}}_e) + I(\mathbf{X}_e; \tilde{\mathbf{X}}_s, \tilde{\mathbf{X}}_e | \mathbf{X}_s) - \beta(I(\mathbf{Y}_s; \tilde{\mathbf{X}}_s, \tilde{\mathbf{X}}_e) + I(\mathbf{Y}_e; \tilde{\mathbf{X}}_s, \tilde{\mathbf{X}}_e | \mathbf{Y}_s)) \\
&= I(\mathbf{X}_s; \tilde{\mathbf{X}}_s, \tilde{\mathbf{X}}_e) + I(\mathbf{X}_e; \tilde{\mathbf{X}}_s, \tilde{\mathbf{X}}_e, \mathbf{X}_s) - I(\mathbf{X}_e; \mathbf{X}_s) - \\
&\quad \beta(I(\mathbf{Y}_s; \tilde{\mathbf{X}}_s, \tilde{\mathbf{X}}_e) + I(\mathbf{Y}_e; \tilde{\mathbf{X}}_s, \tilde{\mathbf{X}}_e, \mathbf{Y}_s) - I(\mathbf{Y}_e; \mathbf{Y}_s)) \\
&= I(\mathbf{X}_s; \tilde{\mathbf{X}}_s, \tilde{\mathbf{X}}_e) + I(\mathbf{X}_e; \tilde{\mathbf{X}}_s, \tilde{\mathbf{X}}_e) - \beta(I(\mathbf{Y}_s; \tilde{\mathbf{X}}_s, \tilde{\mathbf{X}}_e) + I(\mathbf{Y}_e; \tilde{\mathbf{X}}_s, \tilde{\mathbf{X}}_e)) \\
&= I(\mathbf{X}_s; \tilde{\mathbf{X}}_s) + I(\mathbf{X}_s; \tilde{\mathbf{X}}_e | \tilde{\mathbf{X}}_s) + I(\mathbf{X}_e; \tilde{\mathbf{X}}_e) + I(\mathbf{X}_e; \tilde{\mathbf{X}}_s | \tilde{\mathbf{X}}_e) - \\
&\quad \beta(I(\mathbf{Y}_s; \tilde{\mathbf{X}}_s) + I(\mathbf{Y}_s; \tilde{\mathbf{X}}_e | \tilde{\mathbf{X}}_s) + I(\mathbf{Y}_e; \tilde{\mathbf{X}}_e) + I(\mathbf{Y}_e; \tilde{\mathbf{X}}_s | \tilde{\mathbf{X}}_e)) \\
&= I(\mathbf{X}_s; \tilde{\mathbf{X}}_s) + I(\mathbf{X}_s, \tilde{\mathbf{X}}_s; \tilde{\mathbf{X}}_e) - I(\tilde{\mathbf{X}}_s; \tilde{\mathbf{X}}_e) + I(\mathbf{X}_e; \tilde{\mathbf{X}}_e) + I(\mathbf{X}_e, \tilde{\mathbf{X}}_e; \tilde{\mathbf{X}}_s) - I(\tilde{\mathbf{X}}_s; \tilde{\mathbf{X}}_e) - \\
&\quad \beta(I(\mathbf{Y}_s; \tilde{\mathbf{X}}_s) + I(\mathbf{Y}_s, \tilde{\mathbf{X}}_s; \tilde{\mathbf{X}}_e) - I(\tilde{\mathbf{X}}_s; \tilde{\mathbf{X}}_e) + I(\mathbf{Y}_e; \tilde{\mathbf{X}}_e) + I(\mathbf{Y}_e, \tilde{\mathbf{X}}_e; \tilde{\mathbf{X}}_s) - I(\tilde{\mathbf{X}}_s; \tilde{\mathbf{X}}_e)) \\
&= I(\mathbf{X}_s; \tilde{\mathbf{X}}_s) + I(\mathbf{X}_e; \tilde{\mathbf{X}}_e) - \beta I(\mathbf{Y}_s; \tilde{\mathbf{X}}_s) - \beta I(\mathbf{Y}_e; \tilde{\mathbf{X}}_e) \\
&= \gamma_1 I(\mathbf{X}_s; \tilde{\mathbf{X}}_s) + \gamma_2 I(\mathbf{X}_e; \tilde{\mathbf{X}}_e) - \beta\gamma_3 I(\tilde{\mathbf{X}}_s; \mathbf{Y}_s) - \beta\gamma_4 I(\tilde{\mathbf{X}}_e; \mathbf{Y}_e)
\end{aligned}
\tag{15}
$$

The second equation is derived from the $I(\mathbf{X}, \mathbf{Y}; \mathbf{Z}, \mathbf{V}) = I(\mathbf{X}; \mathbf{Z}, \mathbf{V}) + I(\mathbf{Y}; \mathbf{Z}, \mathbf{V} | \mathbf{X})$. The third equation follows from $I(\mathbf{Y}; \mathbf{Z}, \mathbf{V} | \mathbf{X}) = I(\mathbf{Y}; \mathbf{Z}, \mathbf{V}, \mathbf{X}) - I(\mathbf{Y}; \mathbf{X})$. The fourth equation is based

on the orthogonality of external and stable factors. The fifth equation is due to $I(\mathbf{X}; \mathbf{Z}, \mathbf{V}) = I(\mathbf{X}; \mathbf{Z}) + I(\mathbf{X}; \mathbf{V}|\mathbf{Z})$. The sixth equation is derived from $I(\mathbf{X}; \mathbf{V}|\mathbf{Z}) = I(\mathbf{X}, \mathbf{Z}; \mathbf{V}) - I(\mathbf{Z}; \mathbf{V})$, and the seventh equation also relies on the orthogonality of stable and external factors. Without losing generality, we use $\gamma_1$ through $\gamma_4$ to represent them here.

With assumption (a), the equation $I(\mathbf{X}_\mathrm{s}; \tilde{\mathbf{X}}_\mathrm{s}) = I(\tilde{\mathbf{X}}_\mathrm{s}; \mathbf{Y}_\mathrm{s})$ holds, thus

$$\gamma_1 I(\mathbf{X}_\mathrm{s}; \tilde{\mathbf{X}}_\mathrm{s}) - \beta \gamma_3 I(\tilde{\mathbf{X}}_\mathrm{s}; \mathbf{Y}_\mathrm{s}) = (\gamma_1 - \beta \gamma_3) I(\mathbf{X}_\mathrm{s}; \tilde{\mathbf{X}}_\mathrm{s}) \tag{16}$$

With assumption (b), we have,

$$I(\tilde{\mathbf{X}}_\mathrm{e}; \mathbf{Y}_\mathrm{e}) = 0 \tag{17}$$

Plugging equation 16 and 17 into 3, the original minimization objective is equivalent to:

$$\min_{\tilde{\mathcal{D}}} (\gamma_1 - \beta \gamma_3) I(\mathbf{X}_\mathrm{s}; \tilde{\mathbf{X}}_\mathrm{s}) + \gamma_2 I(\mathbf{X}_\mathrm{e}; \tilde{\mathbf{X}}_\mathrm{e}) \tag{18}$$

When $\gamma_1 - \beta \gamma_3 < 0$, $I(\mathbf{X}_\mathrm{s}; \tilde{\mathbf{X}}_\mathrm{s})$ is maximized, effectively rendering $\tilde{\mathbf{X}}_\mathrm{s} \simeq \mathbf{X}_\mathrm{s}$. Meanwhile, as $\gamma_2 > 0$, $I(\mathbf{X}_\mathrm{e}; \tilde{\mathbf{X}}_\mathrm{e})$ is minimized, resulting in $\tilde{\mathbf{X}}_\mathrm{e} \not\sim \mathbf{X}_\mathrm{e}$. $\qquad \square$

### A.2 PROOF OF GENERALIZATION BOUND

*Proof.* Before delving into the proof process, we first introduce the definition of Rademacher complexity and McDiarmid's Inequality:

**Definition A.1 (Rademacher complexity (Mohri, 2018))** *Given a space Z and a fixed distribution $\mathcal{D}$ defined on Z, let $S = z_1, \ldots, z_n$ be a set of examples drawn from i.i.d. from $\mathcal{D}$. Furthermore, let $\mathcal{F}$ be a class of functions $f : Z \to \mathbb{R}$, the empirical Rademacher complexity of $\mathcal{F}$ is defined to be:*

$$\hat{\mathcal{R}}_n(\mathcal{F}) = \mathbb{E}_\sigma \left[ \sup_{f \in \mathcal{F}} \left( \frac{1}{n} \sum_{i=1}^n \sigma_i f(z_i) \right) \right] \tag{19}$$

*where $\sigma_1, \ldots, \sigma_n$ are independent random variables uniform chosen from {-1,1}. The Rademacher complexity of $\mathcal{F}$ is defined as:*

$$\mathcal{R}_n(\mathcal{F}) = \mathbb{E}_\mathcal{D} \left[ \hat{\mathcal{R}}_n(\mathcal{F}) \right] \tag{20}$$

**Theorem A.2 (McDiarmid's Inequality (Mohri, 2018))** *Let $X_1, \cdots, X_n$ be independent random variables, all taking values in the set $\mathcal{X}$. Let f: $\mathcal{X}_1 \times \cdots \times \mathcal{X}_n \to \mathbb{R}$ be any function with the $(c_1, \ldots, c_n)$-bounded difference property: $\forall i, \forall (x_1, \ldots, x_n), x_i' \in \mathcal{X}$, we have $|f(x_1, \ldots, x_i, \ldots, x_n) - f(x_1, \ldots, x_i', \ldots, x_n)| \leq c_i$. Then for any $\epsilon > 0$,*

$$\mathbb{P}\left( f(X_1, \cdots, X_n) - \mathbb{E}[f(X_1, \cdots, X_n)] \geq \epsilon \right) \leq \exp \left( -\frac{2\epsilon^2}{\sum_{i=1}^n c_i^2} \right) \tag{21}$$

Now, let's delve into the proof. In the CDIB framework, by introducing the $i$-th user feature $\Gamma_i$, we generate multiple distributions $\tilde{\mathcal{D}}_i(\Gamma)$ and combine them into a set $\tilde{\mathcal{D}}$. Next, we denote $\Phi(\hat{\mathcal{D}}(\Gamma)) = \sup_{f \in \mathcal{F}} \mathbb{E}_\mathcal{D}[\ell(f; \mathbf{Y})] - \hat{\mathbb{E}}_{\hat{\mathcal{D}}}[\ell(f; \mathbf{Y})]$, and we have:

$$\Phi(\tilde{\mathcal{D}}_i(\Gamma)) - \Phi(\mathcal{D}_{tr}(\Gamma)) \leq \sup_{f \in \mathcal{F}} \hat{\mathbb{E}}_{\mathcal{D}_{tr}}[\ell(f; \mathbf{Y})] - \hat{\mathbb{E}}_{\tilde{\mathcal{D}}_i(\Gamma)}[\ell(f; \mathbf{Y})]$$

$$= \sup_{f \in \mathcal{F}} \frac{f(\dot{\mathbf{h}}, y) - f(\ddot{\mathbf{h}}, y)}{m} \leq \frac{M_2 - M_1}{m} \tag{22}$$

The last inequality holds because CDIB is also trained on $\tilde{\mathcal{D}}$. Since $\Phi$ satisfies the bounded difference property, we can apply the McDiarmid's Inequality to find:

$$\mathbb{P}\left( \Phi(\mathcal{D}_{tr}) - \mathbb{E}_{\mathcal{D}_{tr}}[\Phi(\mathcal{D}_{tr})] \geq \epsilon \right) \leq \exp \left( -\frac{2\epsilon^2}{\sum_{i=1}^m \left( \frac{M_2 - M_1}{m} \right)^2} \right) = \exp \left( -\frac{2\epsilon^2 m}{(M_2 - M_1)^2} \right) \tag{23}$$

Setting the above probability to be less than $\delta$ (*i.e.*, $\exp\left(-\frac{2\epsilon^2 m}{(M_2-M_1)^2}\right) = \delta$), we can solve that $\epsilon = (M_2 - M_1)\sqrt{\frac{\log\frac{2}{\delta}}{m}}$, and we have determined that with probability at least $1 - \delta$:

$$\Phi(\mathcal{D}_{tr}) \leq \mathbb{E}_{\mathcal{D}_{tr}}[\Phi(\mathcal{D}_{tr})] + (M_2 - M_1)\sqrt{\frac{\log\frac{2}{\delta}}{m}} \tag{24}$$

Since $\mathbf{E}_{\mathcal{D}'}[\hat{\mathbf{E}}_{\mathcal{D}'}[\ell(f;\mathbf{Y})]|\mathcal{D}_{tr}] = \mathbf{E}_{\mathcal{D}}[\ell(f;\mathbf{Y})]$ and $\mathbf{E}_{\mathcal{D}'}[\hat{\mathbf{E}}_{\mathcal{D}_{tr}}[\ell(f;\mathbf{Y})]|\mathcal{D}_{tr}] = \hat{\mathbf{E}}_{\mathcal{D}_{tr}}[\ell(f;\mathbf{Y})]$, where $\mathcal{D}'$ is a "ghost sample" independently drawn identically to $\mathcal{D}_{tr}$, we can rewrite the expectation:

$$\begin{aligned}
\mathbb{E}_{\mathcal{D}_{tr}}[\Phi(\mathcal{D}_{tr})] &= \mathbb{E}_{\mathcal{D}_{tr}}\left[\sup_{f\in\mathcal{F}}\mathbb{E}_{\mathcal{D}}[\ell(f;\mathbf{Y})] - \hat{\mathbb{E}}_{\mathcal{D}_{tr}}[\ell(f;\mathbf{Y})]\right] \\
&= \mathbb{E}_{\mathcal{D}_{tr}}\left[\sup_{f\in\mathcal{F}}\mathbb{E}_{\mathcal{D}'}\left(\hat{\mathbb{E}}_{\mathcal{D}'}[\ell(f;\mathbf{Y})] - \hat{\mathbb{E}}_{\mathcal{D}_{tr}}[\ell(f;\mathbf{Y})]\right)\right]
\end{aligned} \tag{25}$$

Since $sup$ is a convex function, we can apply Jensen's Inequality to move the sup inside the expectation:

$$\mathbb{E}_{\mathcal{D}_{tr}}\left[\sup_{f\in\mathcal{F}}\mathbb{E}_{\mathcal{D}'}\left(\hat{\mathbb{E}}_{\mathcal{D}'}[\ell(f;\mathbf{Y})] - \hat{\mathbb{E}}_{\mathcal{D}_{tr}}[\ell(f;\mathbf{Y})]\right)\right] \leq \mathbb{E}_{\mathcal{D}_{tr},\mathcal{D}'}\left[\sup_{f\in\mathcal{F}}\hat{\mathbb{E}}_{\mathcal{D}'}[\ell(f;\mathbf{Y})] - \hat{\mathbb{E}}_{\mathcal{D}_{tr}}[\ell(f;\mathbf{Y})]\right] \tag{26}$$

Multiplying each term in the summation by a Rademacher variable $\sigma_i$ will not change the expectation since $\mathbb{E}[\sigma_i] = 0$. Furthermore, negating a Rademacher variable does not change its distribution. Combining these two facts,

$$\begin{aligned}
&\mathbb{E}_{\mathcal{D}_{tr},\mathcal{D}'}\left[\sup_{f\in\mathcal{F}}\hat{\mathbb{E}}_{\mathcal{D}'}[\ell(f;\mathbf{Y})] - \hat{\mathbb{E}}_{\mathcal{D}_{tr}}[\ell(f;\mathbf{Y})]\right] \\
&= \mathbb{E}_{\mathcal{D}_{tr},\mathcal{D}'}\left[\sup_{f\in\mathcal{F}}\frac{1}{n}\sum_{i=1}^{n}(f(\mathbf{h}',y) - f(\mathbf{h}^o,y))\right] \\
&= \mathbb{E}_{\sigma,\mathcal{D}_{tr},\mathcal{D}'}\left[\sup_{f\in\mathcal{F}}\frac{1}{n}\sum_{i=1}^{n}\sigma_i(f(\mathbf{h}',y) - f(\mathbf{h}^o,y))\right] \\
&\leq \mathbb{E}_{\sigma,\mathcal{D}'}\left[\sup_{f\in\mathcal{F}}\frac{1}{n}\sum_{i=1}^{n}\sigma_i f(\mathbf{h}',y)\right] + \mathbb{E}_{\sigma,\mathcal{D}_{tr}}\left[\sup_{f\in\mathcal{F}}-\frac{1}{n}\sum_{i=1}^{n}\sigma_i f(\mathbf{h}^o,y)\right] \\
&= 2\mathbb{E}_{\sigma,\mathcal{D}_{tr}}\left[\sup_{f\in\mathcal{F}}\frac{1}{n}\sum_{i=1}^{n}\sigma_i f(\mathbf{h}^o,y)\right] = 2\mathcal{R}_n(\mathcal{F})
\end{aligned} \tag{27}$$

The inequality is due to $\sup(A + B) \leq \sup A + \sup B$. Substituting this bound into inequality 24 gives us exactly the Theorem 3.3. $\qquad\square$

### A.3 PROOF OF LOWER BOUND OF $I(\mathbf{Y};\tilde{\mathcal{D}},\mathbf{\Gamma})$

*Proof.* According to the chain rule of mutual information, the conditional prediction term $I(\mathbf{Y};\tilde{\mathcal{D}}|\mathbf{\Gamma})$ can be decomposed as: $I(\mathbf{Y};\tilde{\mathcal{D}}|\mathbf{\Gamma}) = I(\mathbf{Y};\tilde{\mathcal{D}},\mathbf{\Gamma}) - I(\mathbf{Y};\mathbf{\Gamma})$. Intuitively, minimizing the $I(\mathbf{Y};\mathbf{\Gamma})$ aims to reduce the model's capture of personalized interests, which is harmful to satisfying recommendations. Therefore, we only maximize the $I(\mathbf{Y};\tilde{\mathcal{D}},\mathbf{\Gamma})$. Similar to the derivation process in (Choi & Lee,

2023), we have:

$$
\begin{aligned}
I(\mathbf{Y}; \tilde{\mathcal{D}}, \boldsymbol{\Gamma}) &= \mathbb{E}_{\mathbf{Y}, \tilde{\mathcal{D}}, \boldsymbol{\Gamma}} \left[ \log \frac{p(\mathbf{Y} \mid \tilde{\mathcal{D}}, \boldsymbol{\Gamma})}{p(\mathbf{Y})} \right] \\
&= \mathbb{E}_{\mathbf{Y}, \tilde{\mathcal{D}}, \boldsymbol{\Gamma}} \left[ \log \frac{p_{\theta_3}(\mathbf{Y} \mid \tilde{\mathcal{D}}, \boldsymbol{\Gamma})}{p(\mathbf{Y})} \right] + \mathbb{E}_{\tilde{\mathcal{D}}, \boldsymbol{\Gamma}} \left[ D_{KL}(p(\mathbf{Y} \mid \tilde{\mathcal{D}}, \boldsymbol{\Gamma}) \parallel p_{\theta_3}(\mathbf{Y} \mid \tilde{\mathcal{D}}, \boldsymbol{\Gamma}) \right] \\
&\geq \mathbb{E}_{\mathbf{Y}, \tilde{\mathcal{D}}, \boldsymbol{\Gamma}} \left[ \log \frac{p_{\theta_3}(\mathbf{Y} \mid \tilde{\mathcal{D}}, \boldsymbol{\Gamma})}{p(\mathbf{Y})} \right] \\
&= \mathbb{E}_{\mathbf{Y}, \tilde{\mathcal{D}}, \boldsymbol{\Gamma}} \left[ \log p_{\theta_3}(\mathbf{Y} \mid \tilde{\mathcal{D}}, \boldsymbol{\Gamma}) \right] + \mathcal{H}(\mathbf{Y}) \\
&\geq \mathbb{E}_{\mathbf{Y}, \tilde{\mathcal{D}}, \boldsymbol{\Gamma}} \left[ \log p_{\theta_3}(\mathbf{Y} \mid \tilde{\mathcal{D}}, \boldsymbol{\Gamma}) \right]
\end{aligned}
\tag{28}
$$

where $\mathcal{H}(\mathbf{Y})$ represents the entropy of $\mathbf{Y}$ and $\log p_{\theta_3}(\mathbf{Y} | \tilde{\mathcal{D}}, \boldsymbol{\Gamma})$ denotes the variational approximation of $\log p(\mathbf{Y} | \tilde{\mathcal{D}}, \boldsymbol{\Gamma})$. The first and second inequalities hold because of the non-negative inherent in KL-divergence and entropy. So, the lower bound of $I(\mathbf{Y}; \tilde{\mathcal{D}}, \boldsymbol{\Gamma})$ is $\mathbb{E}_{\mathbf{Y}, \tilde{\mathcal{D}}, \boldsymbol{\Gamma}}[\log p_{\theta_3}(\mathbf{Y} | \tilde{\mathcal{D}}, \boldsymbol{\Gamma})]$. $\qquad\square$

### A.4 Proof of $I(\mathcal{D}_{tr}; \tilde{\mathcal{D}}, \boldsymbol{\Gamma}) = I(\mathcal{D}_{tr}; \tilde{\mathcal{D}})$

*Proof.* $\mathcal{D}_{tr}$ is training distribution, $\tilde{\mathcal{D}}$ is the generated distribution, and $\boldsymbol{\Gamma}$ is user features. Remind that $\tilde{\mathcal{D}}$ contains all the information of $\boldsymbol{\Gamma}$ due to its generation process involving $\boldsymbol{\Gamma}$ (see, Equation 7). This implies that $\tilde{\mathcal{D}}$ is a deterministic function of $\boldsymbol{\Gamma}$, *i.e.*, $\boldsymbol{\Gamma} = f(\tilde{\mathcal{D}})$ for some function $f$.

Since $\tilde{\mathcal{D}}$ is a function of $\boldsymbol{\Gamma}$, the joint distribution $p(\dot{\mathbf{h}}_u, \ddot{\mathbf{h}}_u, \mathbf{e}_u)$ can be expressed in terms of $p(\dot{\mathbf{h}}_u, \ddot{\mathbf{h}}_u)$ and the deterministic relationship $\boldsymbol{\Gamma} = f(\tilde{\mathcal{D}})$. Thus, we can write:

$$
p(\dot{\mathbf{h}}_u, \ddot{\mathbf{h}}_u, \mathbf{e}_u) = p(\dot{\mathbf{h}}_u, \ddot{\mathbf{h}}_u) \cdot \delta_{\mathbf{e}_u, f(\ddot{\mathbf{h}}_u)}
\tag{29}
$$

where $\delta$ is the Kronecker delta function, which is 1 if its arguments are equal and 0 otherwise.

The mutual information $I(\mathcal{D}_{tr}; \tilde{\mathcal{D}}, \boldsymbol{\Gamma})$ is given by:

$$
I(\mathcal{D}_{tr}; \tilde{\mathcal{D}}, \boldsymbol{\Gamma}) = \sum_{\dot{\mathbf{h}}_u, \ddot{\mathbf{h}}_u, \mathbf{e}_u} p(\dot{\mathbf{h}}_u, \ddot{\mathbf{h}}_u, \mathbf{e}_u) \log \frac{p(\dot{\mathbf{h}}_u, \ddot{\mathbf{h}}_u, \mathbf{e}_u)}{p(\dot{\mathbf{h}}_u) p(\ddot{\mathbf{h}}_u, \mathbf{e}_u)}
\tag{30}
$$

Given $p(\dot{\mathbf{h}}_u, \ddot{\mathbf{h}}_u, \mathbf{e}_u) = p(\dot{\mathbf{h}}_u, \ddot{\mathbf{h}}_u) \cdot \delta_{\mathbf{e}_u, f(\ddot{\mathbf{h}}_u)}$, we can rewrite the mutual information as:

$$
I(\mathcal{D}_{tr}; \tilde{\mathcal{D}}, \boldsymbol{\Gamma}) = \sum_{\dot{\mathbf{h}}_u, \ddot{\mathbf{h}}_u} p(\dot{\mathbf{h}}_u, \ddot{\mathbf{h}}_u) \cdot \delta_{\mathbf{e}_u, f(\ddot{\mathbf{h}}_u)} \log \frac{p(\dot{\mathbf{h}}_u, \ddot{\mathbf{h}}_u) \cdot \delta_{\mathbf{e}_u, f(\ddot{\mathbf{h}}_u)}}{p(\dot{\mathbf{h}}_u) p(\ddot{\mathbf{h}}_u, \mathbf{e}_u)}
\tag{31}
$$

Since $\delta_{\mathbf{e}_u, f(\ddot{\mathbf{h}}_u)}$ is 1 when $\mathbf{e}_u = f(\ddot{\mathbf{h}}_u)$ and 0 otherwise, the term $\delta_{\mathbf{e}_u, f(\ddot{\mathbf{h}}_u)}$ effectively restricts the summation to the cases where $\mathbf{e}_u = f(\ddot{\mathbf{h}}_u)$. Thus, $p(\ddot{\mathbf{h}}_u, \mathbf{e}_u)$ in the denominator simplifies to $p(\ddot{\mathbf{h}}_u, f(\ddot{\mathbf{h}}_u))$, which is equal to $p(\ddot{\mathbf{h}}_u)$. The mutual information simplifies to:

$$
I(\mathcal{D}_{tr}; \tilde{\mathcal{D}}, \boldsymbol{\Gamma}) = \sum_{\dot{\mathbf{h}}_u} \sum_{\ddot{\mathbf{h}}_u} p(\dot{\mathbf{h}}_u, \ddot{\mathbf{h}}_u) \log \frac{p(\dot{\mathbf{h}}_u, \ddot{\mathbf{h}}_u)}{p(\dot{\mathbf{h}}_u) p(\ddot{\mathbf{h}}_u)}
\tag{32}
$$

This proves that when $\tilde{\mathcal{D}}$ contains all the information of $\boldsymbol{\Gamma}$, the mutual information between $\mathcal{D}_{tr}$ and the joint variables $\tilde{\mathcal{D}}$ and $\boldsymbol{\Gamma}$ is equal to the mutual information between $\mathcal{D}_{tr}$ and $\tilde{\mathcal{D}}$ alone. Intuitively, knowing $\tilde{\mathcal{D}}$ uniquely determines $\boldsymbol{\Gamma}$, $\boldsymbol{\Gamma}$ provide no additional information about $I(\mathcal{D}_{tr}; \tilde{\mathcal{D}}, \boldsymbol{\Gamma})$ beyond what is already known about $\tilde{\mathcal{D}}$. $\qquad\square$

# B    ADDITIONAL EXPERIMENTS

## B.1    CASE STUDY

To verify the CDIB's effectiveness in handling distribution shifts, similar to the case mentioned in the introduction section (see Figure 1), we visualize the interest space learned by CDIB for different users in both the training and testing stages. The result is shown in Figure 7. The result reveals that compared with SASRec, CDIB is less influenced by new trending items (*i.e.*, 634, 8587) or unseen collaboration patterns when modeling the interest of niche users at the testing phase. This indicates the capability of CDIB to capture the users' true interest when faced with distribution shifts.

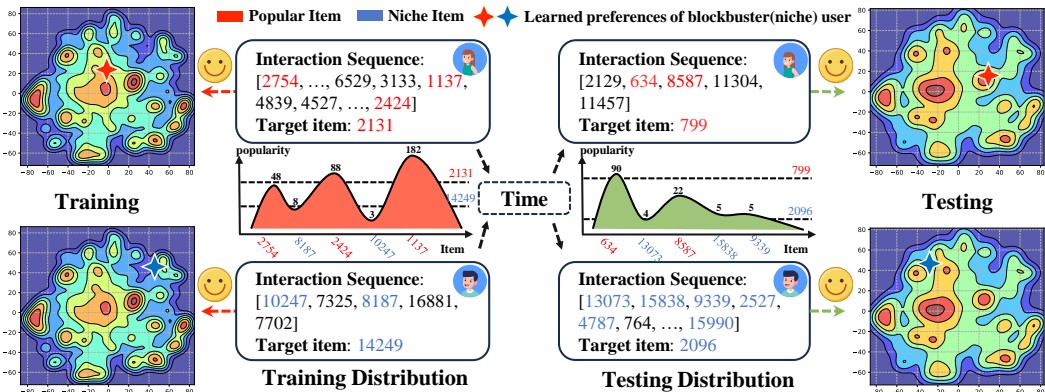

Figure 7: Visualization of the interest space learned by CDIB on Retailrocket dataset.

## B.2    VISUALIZATION OF $\mathbf{M}_u$

To explore the underlying mechanism of the learnable mask, we utilize heatmaps to visualize the popularity of each item in the interaction sequences of six users from the *RetailRocket* dataset, along with the corresponding $1 - \mathbf{M}_u$ obtained by the learnable mask. It is important to note that the higher the $1 - \mathbf{M}_u$ value for an item, the less the model intends to interfere with it, suggesting these items may reflect the user's true interest. The visualization results are displayed in Figure 8. The results show that for niche users (User 509, User 444, User 117), their interaction sequences predominantly feature niche items, which often represent their unique interest. The learnable mask not only protects popular items from being altered but also tends to shield these less common items from interference, as highlighted by the blue dashed box. Conversely, for blockbuster users (User 168, User 107, User 161), who typically interact with trending topics, the learnable mask often identifies these popular items as the users' interest and refrains from modifying them, as indicated by the red dashed box. Compared to traditional hand-crafted data augmentation methods, the learnable mask's ability to adaptively select items for augmentation is crucial. It can intelligently choose which items to enhance based on user features without distorting the original user interest. This approach helps generate more promising augmented samples and, to some extent, avoids introducing extraneous noise.

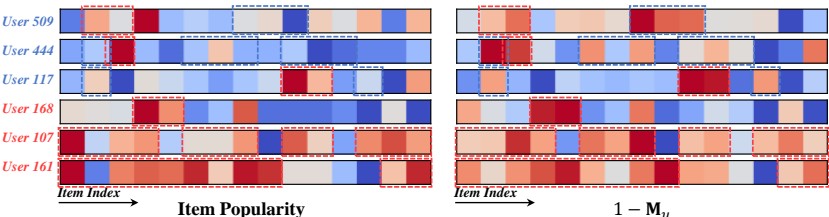

Figure 8: The visualization of interacted items' popularity and the corresponding $1 - \mathbf{M}_u$.

B.3 Visualization of $\tilde{\mathcal{D}}$

We visualize the representation space of some origin and augmented items to explore its inner mechanisms. Specifically, we reduce the dimensionality of items within the users' interaction sequences and those enhanced by latent diffusion using t-SNE (Van der Maaten & Hinton, 2008), then visualize their two-dimensional outcomes. Additionally, considering that $\beta$ is a critical hyperparameter for balancing the diversity and reliability of generated distribution, we also visualized the generation results for different $\beta$ values, as depicted in Figure 9. From the visualization results, it can be concluded that at lower $\beta$ values (*i.e.*, 1e0, 1e1), feature collapse occurs in the generated items. This collapse may happen because a small $\beta$ shifts the model's focus towards generating low similarity distributions (minimizing $I(\mathcal{D}_{tr}; \tilde{\mathcal{D}}|\mathbf{\Gamma})$), thereby neglecting the intrinsic features of the original dataset. Such generated distributions can introduce unnecessary noise to the model, complicating its learning process. Hence, the model performs poorly when $\beta$ is small (see Section 4.2). As $\beta$ increases, the distribution generated by latent diffusion gradually aligns more closely with the original distribution and retains a clustering effect of popular items to some extent, while the overall distribution becomes more uniform, aiding the model's learning process (Yu et al., 2022). When $\beta$ reaches 1e4, there is no distinguishable difference between the generated and original distributions. At this level, the model prioritizes preserving the original data characteristics as much as possible (maximizing $I(\mathbf{Y}; \tilde{\mathcal{D}}|\mathbf{\Gamma})$, which limits the exploration of out-of-distribution scenarios and decreases overall effectiveness.

B.4 Results on Diverse Augmentation Methods

Within the scope of our ablation study, we added Gaussian noise to the original data for augmentation rather than using diffusion-based augmentation techniques. We also conducted experiments with random data augmentation strategies. Specifically, we performed random masking, cropping, and reordering on the original interaction sequences to augment the data. The overall results are presented in Table 2. It can be seen that our diffusion-based augmentation method performs better than the other two techniques.

Table 2: Performance over diverse data augmentation methods.

| Method | MovieLens-100K | | Retailrocket | | Amazon-Beauty | | Amazon-Sports | |
| --- | --- | --- | --- | --- | --- | --- | --- | --- |
| | HitRate $\uparrow$ | NDCG $\uparrow$ | HitRate $\uparrow$ | NDCG $\uparrow$ | HitRate $\uparrow$ | NDCG $\uparrow$ | HitRate $\uparrow$ | NDCG $\uparrow$ |
| random | $11.09_{\pm 0.22}$ | $5.08_{\pm 0.13}$ | $19.84_{\pm 0.17}$ | $8.76_{\pm 0.11}$ | $8.98_{\pm 0.17}$ | $4.42_{\pm 0.05}$ | $4.90_{\pm 0.18}$ | $2.30_{\pm 0.10}$ |
| Gaussian Noise | $11.59_{\pm 0.16}$ | $5.53_{\pm 0.12}$ | $20.82_{\pm 0.16}$ | $9.37_{\pm 0.04}$ | $9.06_{\pm 0.15}$ | $\mathbf{4.60_{\pm 0.06}}$ | $4.92_{\pm 0.21}$ | $\mathbf{2.40_{\pm 0.08}}$ |
| CDIB | $\mathbf{11.89_{\pm 0.16}}$ | $\mathbf{5.67_{\pm 0.09}}$ | $\mathbf{21.12_{\pm 0.14}}$ | $\mathbf{9.41_{\pm 0.10}}$ | $\mathbf{9.17_{\pm 0.04}}$ | $4.56_{\pm 0.04}$ | $\mathbf{4.95_{\pm 0.11}}$ | $2.38_{\pm 0.07}$ |

B.5 Sensitivity Analysis on $\alpha_1$ and $\alpha_2$

We have further added sensitivity analysis of hyperparameters for $\alpha_1$ and $\alpha_2$ on ml-100k and retailrocket, respectively. Specifically, we evaluate our model by varying the $\alpha_1$ and $\alpha_2$ in {0.01, 0.1, 1.0, 5.0, 10.0}, respectively. The results are present in Table 3 and Table 4. We conclude our observations as follows: **(i)** Optimal performance across both datasets is attained at $\alpha_1 = 1.0$, marking a peak in performance that rises to it and then begins to decline. If $\alpha_1$ is set too low, the diffusion model's generative capabilities are diminished, potentially leading to the creation of noise samples that can hinder model training. Conversely, if $\alpha_1$ is too high, the auxiliary task takes precedence in the model's optimization process, which can adversely impact the model's recommendation capabilities. **(ii)** Optimal model performance is consistently achieved at $\alpha_2 = 1.0$ across all datasets, after which there is a notable decline in performance as $\alpha_2$ increases to 10.0, with pronounced effects on the Retail dataset. This decrease may be due to the model's overly focus on model performance on the generated data distribution at higher $\alpha_2$ values, potentially obscuring the model's capacity to extract essential information from the original datasets' distribution.

Table 3: Sensitivity analysis on $\alpha_1$

| $\alpha_1$ | MovieLens-100K | | Retailrocket | |
|---|---|---|---|---|
| | HitRate ↑ | NDCG ↑ | HitRate ↑ | NDCG ↑ |
| 0.01 | $\mathbf{11.91}_{\pm 0.19}$ | $5.58_{\pm 0.13}$ | $20.92_{\pm 0.15}$ | $9.28_{\pm 0.12}$ |
| 0.1 | $11.76_{\pm 0.17}$ | $5.59_{\pm 0.14}$ | $20.96_{\pm 0.15}$ | $9.38_{\pm 0.11}$ |
| 1.0 | $11.89_{\pm 0.16}$ | $\mathbf{5.67}_{\pm 0.09}$ | $\mathbf{21.12}_{\pm 0.14}$ | $9.41_{\pm 0.10}$ |
| 5.0 | $11.16_{\pm 0.14}$ | $5.30_{\pm 0.08}$ | $21.10_{\pm 0.15}$ | $\mathbf{9.44}_{\pm 0.13}$ |
| 10.0 | $10.99_{\pm 0.17}$ | $5.12_{\pm 0.12}$ | $20.98_{\pm 0.12}$ | $9.40_{\pm 0.11}$ |

Table 4: Sensitivity analysis on $\alpha_2$

| $\alpha_2$ | MovieLens-100K | | Retailrocket | |
|---|---|---|---|---|
| | HitRate ↑ | NDCG ↑ | HitRate ↑ | NDCG ↑ |
| 0.01 | $11.11_{\pm 0.18}$ | $4.92_{\pm 0.14}$ | $19.69_{\pm 0.15}$ | $8.58_{\pm 0.12}$ |
| 0.1 | $11.44_{\pm 0.17}$ | $5.19_{\pm 0.11}$ | $20.28_{\pm 0.15}$ | $8.81_{\pm 0.13}$ |
| 1.0 | $\mathbf{11.89}_{\pm 0.16}$ | $\mathbf{5.67}_{\pm 0.09}$ | $\mathbf{21.12}_{\pm 0.14}$ | $\mathbf{9.41}_{\pm 0.10}$ |
| 5.0 | $8.42_{\pm 0.14}$ | $3.81_{\pm 0.07}$ | $16.00_{\pm 0.13}$ | $7.83_{\pm 0.12}$ |
| 10.0 | $5.73_{\pm 0.13}$ | $2.63_{\pm 0.6}$ | $5.13_{\pm 0.15}$ | $2.50_{\pm 0.13}$ |

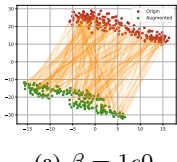 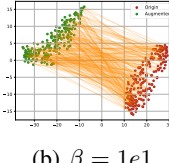 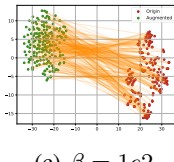 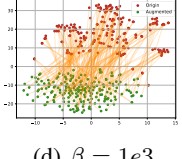 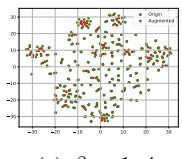

(a) $\beta = 1e0$    (b) $\beta = 1e1$    (c) $\beta = 1e2$    (d) $\beta = 1e3$    (e) $\beta = 1e4$

Figure 9: The visualization of generated distribution *w.r.t* $\beta$.

### B.6 APPROXIMATE PERCENTAGE OF VALUES OF 1 IN $\mathbf{M}_u$

we record the percentage of values of 1 in $\mathbf{M}_u$ before and after adding the $\mathcal{L}_{mask}$, the results are shown in Table 5. As we can see, after the introduction of the $\mathcal{L}_{mask}$, the percentage of values of 1 in $\mathbf{M}_u$ increased from 0.0% to an average of 17.4%.

Table 5: approximate percentage of values of 1 in $\mathbf{M}_u$

| | ML-100K | Retailrocket | Amazon-Beauty | Amazon-Sports |
|---|---|---|---|---|
| w/o $\mathcal{L}_{mask}$ | 0.0% | 0.0% | 0.0% | 0.0% |
| w $\mathcal{L}_{mask}$ | 13.5% | 22.1% | 15.5% | 18.4% |

### B.7 RESULTS ON ML-1M DATASET

To demonstrate the effectiveness of the proposed CDIB, we conducted experiments on the larger ML-1M dataset. The results, presented in Table 6, show that our method achieves the best performance.

Table 6: Performance on ML-1M. The best results and second-best are in **bold** and underline. All the numbers are percentage values with "%" omitted (mean±std). The experiments are conducted 5 times.

| | GRU4Rec | Caser | SASRec | IPS | S-DRO | DROS | DCG | DCF | CL4SRec | DuoRec | DCRec | CDIB |
|---|---|---|---|---|---|---|---|---|---|---|---|---|
| HitRate ↑ | $17.79_{\pm 0.06}$ | $14.84_{\pm 0.14}$ | $18.61_{\pm 0.08}$ | $18.33_{\pm 0.04}$ | $17.92_{\pm 0.09}$ | $19.21_{\pm 0.08}$ | $18.79_{\pm 0.09}$ | $18.81_{\pm 0.11}$ | $18.66_{\pm 0.12}$ | $18.97_{\pm 0.06}$ | $18.11_{\pm 0.10}$ | $20.57_{\pm 0.07}$ |
| NDCG ↑ | $7.43_{\pm 0.03}$ | $5.60_{\pm 0.06}$ | $9.28_{\pm 0.02}$ | $8.98_{\pm 0.02}$ | $7.87_{\pm 0.03}$ | $9.55_{\pm 0.03}$ | $9.41_{\pm 0.03}$ | $9.43_{\pm 0.02}$ | $9.27_{\pm 0.03}$ | $9.34_{\pm 0.04}$ | $9.17_{\pm 0.04}$ | $9.86_{\pm 0.03}$ |

## C IMPLEMENTATION DETAILS

The CDIB model is implemented using Pytorch 1.13.0 (Paszke et al., 2019) and Python 3.8.13. Experiments are conducted using two NVIDIA GeForce RTX 3090 GPUs. To ensure a fair comparison, we adopt the widely used experimental environment RecBole (Zhao et al., 2021). The parameters are initialized using a Gaussian distribution $\mathcal{N}(0, 0.02)$ and optimized with the Adam optimizer at a learning rate of 0.001. Moreover, we adopt the early-stop strategy to train the models and set the maximum sequence length to 50. We run the codes for GRU4Rec, Caser, SASRec, and CL4SRec,

which are reproduced by the RecBole team[3]. We also reproduce the implementations of IPS, S-DRO, DROS, DuoRec, and DCRec to the RecBole environment.

## C.1 ALGORITHM

The pseudo-algorithms for the training and inference stages of CDIB are presented in Algorithm 1 and Algorithm 2, respectively.

---

**Algorithm 1:** The Training Stage of the Proposed CDIB Algorithm

---

**Input:** user set $\mathcal{U} = \{u\}$, item set $\mathcal{I} = \{i\}$, interaction sequences $\mathcal{S}_{tr}$, learning rate $\eta$, and batch size $B$.

**Output:** trained recommender $\xi^*$.

1   Initialize all parameters;
2   **while** *not converge* **do**
3      Sample a batch $\{u\}_1^B$ and $\{s_u\}_1^B$ from $\mathcal{S}_{tr}$
4      Embed users $\{u\}_1^B$ and items $\{s_u\}_1^B$ to get the corresponding embedding $\boldsymbol{\Gamma}$ and $\mathbf{H}_u$
5      # Generating Distribution $\tilde{\mathcal{D}}$
6      Calculate the mask $\mathbf{M}_u$ using the Learnable Mask
7      Mask the stable items by $\mathbf{H}_u^0 = \mathbf{H}_u \odot \mathbf{M}_u$
8      Forward diffusion process: $\mathcal{N}(\mathbf{H}_u^t; \sqrt{1 - \beta_t}\mathbf{H}_u^{t-1}, \beta_t\mathbf{I})$
9      Reverse diffusion process: $p(\mathbf{H}_u^T) \prod_{t=1}^T p_{\theta_2}(\mathbf{H}_u^{t-1}|\mathbf{H}_u^t)$
10     Sample sensitive items to get diverse data $\tilde{\mathbf{H}}_u = \tilde{\mathbf{H}}_u^0 + \mathbf{H}_u \odot (1 - \mathbf{M}_u)$
11     Calculate the $\mathcal{L}_{gd} = \mathcal{L}_{con} + \mathcal{L}_{mask}$
12     # Optimizing with CDIB
13     Encode origin and augmented interaction sequence using Transformer Recommender
14     Obtain the overall loss: $\mathcal{L}_{total} = \mathcal{L}_{pred} + \alpha_1\mathcal{L}_{gd} + \alpha_2(\beta\mathcal{L}_{reg} + \mathcal{L}_{gen})$
15     Update $\xi$ to minimize $\mathcal{L}_{total}$
16   **end**
17   **return** trained recommender $\xi^*$

---

---

**Algorithm 2:** The testing Stage of the Proposed CDIB Algorithm

---

**Input:** interaction sequences $\mathcal{S}_{te}$, trained recommender $\xi^*$, and item set $\mathcal{I}$.

**Output:** recommended items.

1   **for** $s_u \in \mathcal{S}_{te}$ **do**
2      **return** $\arg\max_{i \in \mathcal{I}} p(i|\xi^*(s_u))$
3   **end**

---

## C.2 HYPERPARAMETER

We tune the hyperparameters as follows: Batch Size $\in \{64, 128, 256, 512, 1024\}$; Dropout Rate $\in \{0.1, 0.3, 0.5, 0.7\}$; $\beta \in \{10, 100, 1000, 10000\}$. For a fair comparison, we standardized all common hyperparameters across models and configured the unique hyperparameters according to the settings provided by the corresponding authors. The hyperparameters settings of CDIB are present in Table 7.

## C.3 DETAILS ABOUT $\mathcal{L}_{pred}$ AND $\mathcal{L}_{reg}$

$\mathcal{L}_{pred}$ is a negative log-likelihood function of the expected next item $i_{L+1}$ of an origin interaction sequence $s_u$, where we adopt cross-entropy loss under the full set of items:

$$\mathcal{L}_{pred} = \frac{1}{|\mathcal{U}|} \sum_{u=1}^{|\mathcal{U}|} -\log\left(\frac{\exp(\dot{\mathbf{h}}_u \cdot \mathbf{e}_u^{L+1})}{\sum_{i \in \mathcal{I}} \exp(\dot{\mathbf{h}}_u \cdot \mathbf{e}_i)}\right) \tag{33}$$

---

[3]https://github.com/RUCAIBox/RecBole/tree/master/recbole/model/sequential_recommender

Table 7: Hyperparameter specifications

| Dataset | ML-100K | Retailrocket | Amazon-Beauty | Amazon-Sports |
|---|---|---|---|---|
| Optimizer | Adam | Adam | Adam | Adam |
| Batch Size | 512 | 512 | 512 | 512 |
| Learning Rate | 0.001 | 0.001 | 0.001 | 0.001 |
| Embedding Size | 64 | 64 | 64 | 64 |
| Hidden Size | 256 | 256 | 256 | 256 |
| Dropout Rate | 0.5 | 0.5 | 0.5 | 0.5 |
| Temperature $\tau$ | 1.0 | 1.0 | 1.0 | 1.0 |
| Lagrange multiplier $\beta$ | 1000 | 1000 | 1000 | 10000 |

where $\dot{\mathbf{h}}_u$ is the representation of the origin interaction sequence, $\mathbf{e}_u^{L+1}$ is the embedding of the next interacted item, and $\mathbf{e}_i$ is the embedding of item $i$.

$\mathcal{L}_{reg}$ is the regularization loss, which encourages that essential information to the target item $i_{L+1}$ is preserved, where we utilized the cross entropy loss under the full set of items:

$$\mathcal{L}_{reg} = \frac{1}{|\mathcal{U}|} \sum_{u=1}^{|\mathcal{U}|} -\log \left( \frac{\exp(\ddot{\mathbf{h}}_u \cdot \mathbf{e}_u^{L+1})}{\sum_{i \in \mathcal{I}} \exp(\ddot{\mathbf{h}}_u \cdot \mathbf{e}_i)} \right) \tag{34}$$

where $\ddot{\mathbf{h}}_u$ is the representation of the generated interaction sequence, $\mathbf{e}_u^{L+1}$ is the embedding of the next interacted item, and $\mathbf{e}_i$ is the embedding of item $i$.

## D COMPLEXITY ANALYSES

**Time Complexity Analysis.** For one batch of training data, the computational cost of the learnable mask is $\mathcal{O}(4Ld)$, and the distribution generator is $\mathcal{O}(Ld + TLd^2)$, which means the time cost of generating new distribution by CDIB is $\mathcal{O}(5Ld + TLd^2)$ in the training stage. With the attention calculations, the time complexity of the transformer recommender is $\mathcal{O}(2L^2hd + Lhd^2)$, which is also the total time cost in the testing stage. Moreover, the time complexity to compute $\mathcal{L}_{gd}$ is $\mathcal{O}(L(2d+1))$, $\mathcal{L}_{reg}$ is $\mathcal{O}(|\mathcal{I}|d)$, $\mathcal{L}_{gen}$ is $\mathcal{O}(2|\mathcal{U}|d)$, and $\mathcal{L}_{pred}$ is $\mathcal{O}(|\mathcal{I}|d)$, therefore, the total time cost to compute the loss is $\mathcal{O}(L(2d+1) + 2|\mathcal{I}|d + 2|\mathcal{U}|d)$. Consider that $|\mathcal{I}|$ and $|\mathcal{U}|$ are much larger than $L$ empirically, the overall time complexity of CDIB is $\mathcal{O}((|\mathcal{I}| + |\mathcal{U}|)d)$, which is affordable.

**Space Complexity Analysis.** Compared to the naive sequential recommendation model SAS-Rec (Kang & McAuley, 2018), our CDIB uses extra parameters costs $\mathcal{O}(|\mathcal{B}|d)$ to represent the users' embeddings, which only occur in the training stage, and $\mathcal{O}(d^2)$ to develop the learnable mask and distribution generator, which is affordable. The running time and model size in the training stage are shown in Table 8, and that of the testing stage is present in Table 9, where also present the performance improvement compared with SASRec.

Table 8: Running time and Model size at the training stage

| Dataset | SASRec | | DROS | | DCRec | | CDIB | |
|---|---|---|---|---|---|---|---|---|
| | Running Time | Model Size | Running Time | Model Size | Running Time | Model Size | Running Time | Model Size |
| ML-100K | 00h 15m 02s | 0.21 M | 01h 48m 01s | 0.21 M | 01h 26m 55s | 0.22 M | 00h 37m 21s | 0.42 M |
| Retail | 01h 02m 32s | 1.24 M | 02h 25m 38s | 1.24 M | 06h 36m 12s | 1.25 M | 01h 19m 24s | 2.80 M |
| Beauty | 01h 03m 39s | 0.87 M | 01h 39m 34s | 0.87 M | 02h 11m 21s | 0.89 M | 01h 08m 40s | 2.45 M |
| Sports | 01h 34m 48s | 1.27 M | 03h 35m 04s | 1.27 M | 02h 42m 11s | 1.29 M | 00h 50m 38s | 3.70 M |

Table 9: Running time, Model size and Performance Improvement at the testing stage

| Dataset | SASRec | | DROS | | | DCRec | | | CDIB | | |
|---|---|---|---|---|---|---|---|---|---|---|---|
| | Running Time | Model Size | Running Time | Model Size | Performance Improvement | Running Time | Model Size | Performance Improvement | Running Time | Model Size | Performance Improvement |
| ML-100K | 0.13s | 0.21 M | 0.12s | 0.21 M | ↑ 8.10% | 0.14s | 0.22 M | ↓ 6.94% | 0.12s | 0.21 M | ↑ 17.23% |
| Retailrocket | 0.37s | 1.24 M | 0.45s | 1.24 M | ↓ 0.32% | 0.43s | 1.25 M | ↑ 1.72% | 0.38s | 1.24 M | ↑ 8.53% |
| Beauty | 0.34s | 0.87 M | 0.32s | 0.87 M | ↓ 3.54% | 0.38s | 0.89 M | ↓ 9.13% | 0.31s | 0.87 M | ↑ 6.34% |
| Sports | 0.47s | 1.27 M | 0.48s | 1.27 M | ↑ 4.41% | 0.58s | 1.29 M | ↓ 11.34% | 0.48s | 1.27 M | ↑ 7.29% |

# E  DATASETS, BASELINES, METRICS, AND RELATED WORKS

## E.1  DETAILED DATASETS DESCRIPTION

**ML-100K.** ML-100K is sourced from MovieLens[4], a recommendation system and virtual community website established by the GroupLens Research Project at the University of Minnesota's School of Computer Science and Engineering.

**Retailrocket.** The Retailrocket dataset was collected from a real-world e-commerce website[5]. We utilize viewing sequences to train and test our model. Following the approach in (Yang et al., 2023b), we exclude items interacted with fewer than five times to mitigate the cold-start issue. Additionally, sequences shorter than three interactions are removed.

**Amazon.** The Amazon-Beauty and Amazon-Sports datasets compile user-item interactions from Amazon[6] in the Beauty and Sports product categories, respectively. Employing preprocessing similar to that used for Retailrocket, we filter out items with fewer than five interactions and sequences shorter than three. The statistics of our evaluation datasets are detailed in Table 10.

Table 10: Dataset statics

| Dataset | ML-100K | Retailrocket | Amazon-Beauty | Amazon-Sports | ML-1M |
|---|---|---|---|---|---|
| #Users | 944 | 22179 | 22364 | 35599 | 6041 |
| #Items | 1683 | 17804 | 12102 | 18358 | 3417 |
| #Interactions | 100000 | 240938 | 198502 | 296337 | 999611 |
| Avg. actions of users | 106.04 | 10.86 | 8.87 | 8.32 | 165.50 |
| Avg. actions of items | 59.45 | 13.53 | 16.40 | 16.14 | 292.63 |
| Sparsity | 93.71% | 99.94% | 99.93% | 99.95% | 95.16% |

## E.2  DETAILED BASELINES DESCRIPTION

We compare CDIB with ten methods from diverse research lines, covering:

1. **Naive Sequential Recommendation Methods**: These methods have been effective techniques to capture the evolving pattern of users' interest.

   - **GRU4Rec** (Hidasi et al., 2016): GRU4Rec utilizes the Gated Recurrent Unit (GRU) for session-based recommendations, providing strong sequence modelling capabilities.
   - **Caser** (Tang & Wang, 2018): Caser is a CNN-based approach that employs horizontal and vertical convolutional filters to capture sequential patterns.
   - **SASRec** (Kang & McAuley, 2018): SASRec applies a multi-head self-attention mechanism to encode item-wise sequential correlations, suitable for long sequence data.

2. **Reweighting Methods**: These methods aim to develop a more unbiased and robust model by adjusting the weight of each training instance.

   - **IPS** (Schnabel et al., 2010): IPS re-weights each training instance with inverse popularity score to eliminate popularity bias.

---

[4]https://movielens.org/

[5]https://www.kaggle.com/datasets/retailrocket/ecommerce-dataset/

[6]https://www.amazon.com/

3. **DRO Methods**: DRO Methods integrate the distributionally robust optimization (Hu & Hong, 2013) to the sequential recommendation to obtain a recommender with better generalization ability.

   - **S-DRO** (Wen et al., 2022): This model adds streaming optimization improvement to the Distributionally Robust Optimization (DRO) framework to mitigate the amplification of Empirical Risk Minimization (ERM) on popularity bias.
   - **DROS** (Yang et al., 2023b): It introduces a carefully designed distribution adaption paradigm, which considers the dynamics of data distribution and explores possible distribution shifts between training and testing.

4. **Diffusion-based Augmentation Methods**: Diffusion-based Augmentation Methods utilize diffusion technical to enrich the sparse training data to improve the model performance.

   - **DiffuASR** (Liu et al., 2023): This model designs a Sequential U-Net to capture sequence information while predicting the added noise. Additionally, two guiding strategies (DiffuASR-CG and DiffuASR-CF) are implemented to steer DiffuASR, ensuring it generates items that align more closely with the preferences in the original sequence.

5. **Contrastive Learning Methods**: CL methods adopt data augmentation to enhance the robustness of recommenders.

   - **CL4SRec** (Xie et al., 2022): CL4SRec employs random corruption techniques like cropping, masking, and reordering to generate contrastive views.
   - **DuoRec** (Qiu et al., 2022): DuoRec introduces supervised positive sampling to obtain high-quality positive pairs.
   - **DCRec** (Yang et al., 2023a): DCRec unifies sequential pattern encoding with global collaborative relation modelling through adaptive conformity-aware augmentation.

### E.3 DETAILED METRICS DESCRIPTION

We focus on top-N item recommendations and utilize two widely used metrics for evaluation: Hit Rate (HR)@$N$ and Normalized Discounted Cumulative Gain (NDCG)@$N$. These metrics are crucial for assessing the recommendation accuracy at the top-$N$ ranked positions (Kang & McAuley, 2018; Yang et al., 2023a; Xia et al., 2023). The models are evaluated using an all-ranking protocol (He et al., 2020), which provides a robust and comprehensive performance assessment. The metrics are formally calculated as follows:

$$HR@N = \frac{\sum_{i=1}^{M} \sum_{j=1}^{N} r_{i,j}}{M}; \quad NDCG@N = \sum_{i=1}^{M} \frac{\sum_{j=1}^{N} r_{i,j} / \log_2(j+1)}{M \cdot IDCG_i} \tag{35}$$

where $M$ denotes the number of tested users, $r_{i,j} = 1$ if the $j$-th item in the ranked list for the $i$-th user is positive, and $r_{i,j} = 0$ otherwise. The numerator of $NDCG@N$ is the discounted cumulative gain (DCG) at $N$, and $IDCG_i$ is the ideal maximum $DCG@N$ value for the $i$-th tested user.

### E.4 MORE RELATED WORKS

**Sequential Recommendation** is designed to predict the next item a user is likely to prefer based on their interaction history. Traditional methods have leveraged Markov chains to capture first-order item-to-item correlations through transition matrices (Rendle et al., 2010; He & McAuley, 2016). With the development of deep learning, which excels at modeling complex sequential patterns, various deep recommendation models have been developed. For instance, GRU4Rec (Hidasi et al., 2016) employs Gated Recurrent Unit (GRU) units to model the temporal dynamics of interaction sequences. Caser (Tang & Wang, 2018) uses a time convolutional neural network (TCN) to account for both long-term and short-term user interests in personalized recommendations. SASRec (Kang & McAuley, 2018) and BERT4Rec (Sun et al., 2019) enhance computational efficiency in lengthy sequences by incorporating self-attention mechanisms. More recently, inspired by selective state space models (Gu & Dao, 2024), Mamba4Rec (Liu et al., 2024a) has been introduced, utilizing the mamba framework to recommend items efficiently. Despite their capabilities, these models often suffer performance declines when OOD occurs. To address this, CDIB introduces a user feature-guided generation approach that proactively explores OOD scenarios during the training phase, enhancing the model's generalization capabilities.

**Distributionally Robust Sequential Recommendation** has recently attracted significant research interest, which aims to train a model that performs well not only at the training stage but also at the testing stage. Methods like reweighting and DRO (Schnabel et al., 2010; Bottou et al., 2013; Wang et al., 2022b; Yang et al., 2023b; Wen et al., 2022) presume that the test dataset's distribution can be inferred from prior knowledge. For example, IPS (Schnabel et al., 2010) re-weight each instance with the inverse propensity score, which implicitly assumes the testing distribution is uniform (Zhang et al., 2023). DROS (Yang et al., 2023b) unifies the DRO and sequential recommendation paradigms to enhance model robustness against distribution shifts but faces challenges with sparse data. Causal inference methods capture real causal relationships but assume the causal graph is static (Wang et al., 2023b; He et al., 2022; Yang et al., 2020; Wang et al., 2022a), while contrastive learning approaches seek to enrich the training data distribution through data augmentation (Liu et al., 2021; Xie et al., 2022; Yang et al., 2023a; Qiu et al., 2022; Zhao et al., 2023), but hardly rely on the data augmentation strategies. What's more, most of the existing models ignore the user's sensitivity during the process of distribution shift. To fill the gap, we introduce the CDIB principle, using the user features to guide the exploration of the other distribution.

**Information Bottleneck with Conditional Information** has been increasingly utilized in recent research. Various studies have adopted the information bottleneck (IB) principle by incorporating conditional, aiming to extract information that aligns with specific objectives. The conditional information bottleneck (CIB) theory (Gondek & Hofmann, 2003) has been applied in methods such as CGIB (Lee et al., 2023) to identify crucial molecular structures that predict interactions between graph pairs, with a focus on significant subgraphs. TimeCIB (Choi & Lee, 2023) extends the CIB to time series data imputation, ensuring the preservation of essential temporal information. Drawing inspiration from these precedents, CDIB employs CIB to steer the generation of distributions, enhancing the model's robustness. To the best of our knowledge, CDIB is the first application of CIB to guide the distribution generation process.

**Diffusion-based Augmentation Models** Earlier approaches like Diff4Rec (Wu et al., 2023) and DiffuASR (Liu et al., 2023) followed a three-step process: training the diffusion model, generating new data with the diffusion model, and then training the recommendation model on new data, which can lead to a disconnect between the generation and downstream tasks due to the discrete nature of these stages, preventing the flow of gradient information. Our model, however, employs an end-to-end training approach, which maintains the alignment between the generation and downstream tasks. What's more, in the SR scenario, interaction data is very sensitive (Ye et al., 2023), and there is a risk of losing significant information during the data augmentation phase, which may compromise the quality of the generated data, a concern overlooked in previous methods. Our model addresses this by utilizing a learnable mask mechanism to safeguard critical interactions adaptively and is guided by IB theory in the generation process.

# F  MECHANISM OF THE DIFFUSION MODEL

**Diffusion Models** have shown impressive generative performance across several domains, including computer vision (Ho et al., 2020; Rombach et al., 2022), natural language processing (Austin et al., 2021), and time series (Rasul et al., 2021; Shen & Kwok, 2023; Liu et al., 2024b). Generally, diffusion models consist of two pivotal phases: the *forward process* and the *reverse process*. In the *forward process*, Gaussian noise is incrementally introduced into the initial data sample $x_0 \sim q(x_0)$, creating a sequence $x_{1:T}$ through $T$ steps in a Markov chain, which can be formulated as follows:

$$q(x_t|x_{t-1}) = \mathcal{N}(x_t; \sqrt{1 - \beta_t}x_{t-1}, \beta_t \mathbf{I}), \tag{36}$$

where $\mathcal{N}$ indicates the Gaussian distribution and $\beta_t \in (0, 1)$ specifies the scale of noise introduced at each step $t$. Through the reparameterization trick and principle that the sum of two independent Gaussian noises is also Gaussian, $x_t$ can be directly derived from $x_0$ as $x_t = \sqrt{\bar{\alpha}_t}x_0 + \sqrt{1 - \bar{\alpha}_t}\epsilon_t$, with $\epsilon_t \sim \mathcal{N}(0, \mathbf{I})$ as the added noise, and $\bar{\alpha}_t = \prod_{t'=1}^{T}(1 - \beta_{t'})$ signifying the cumulative product of noise scaling factors. The *reverse process*, conversely, aims to iteratively remove the noise from $x_t$ to reconstruct $x_{t-1}$ and ultimately retrieve the original sample $x_0$. This is formulated as:

$$p_\theta(x_{0:T}) = p(x_T)\prod_{t=1}^{T} p_\theta(x_{t-1}|x_t) \tag{37}$$

where $p(x_T) \sim \mathcal{N}(0, \mathbf{I})$ and $\prod_{t=T}^{1} p_\theta(x_{t-1}|x_t)$ denotes the process of sequentially deducing $x_{t-1}$ by reversing the estimated Gaussian noise from $x_t$ via a neural network parameterized by $\theta$. The

diffusion model's learning objective is thereby distilled to:

$$\mathcal{L}_\epsilon = \sum\nolimits_{t=2}^{T} \mathbb{E}_{t,\epsilon}[||\epsilon_t - \epsilon_\theta(x_t, t)||_2^2] \tag{38}$$

where $\epsilon_\theta(x_t, t)$ represents the noise modeled to have been added to $x_{t-1}$ in the forward process. This mechanism enables the diffusion model to approximate complex data distributions and generate high-quality samples, making it particularly effective for tasks requiring diverse and structured data generation.

## G NOTATION TABLE

Table 11: Notation Table

| Notation | Description |
|----------|-------------|
| $\mathbf{e}_u$ | User embedding for user $u$ |
| $\mathbf{h}_u^l$ | Embedding of the $l$-th item interacted by user $u$ |
| $\mathbf{H}_u$ | Hidden representation of user $u$'s interaction sequence |
| $s_u$ | Interaction sequence of user $u$ |
| $\tilde{s}_u$ | Masked interaction sequence |
| $\mathbf{M}_u$ | Learnable mask applied to $s_u$ |
| $\mathbf{H}_u^t$ | Noised hidden representation $\mathbf{H}_u$ at the $t$-th step |
| $\tilde{\mathbf{H}}_u^0$ | Reconstructed hidden representation |
| $\tilde{\mathbf{H}}_u$ | Hidden representation of the generated interaction sequence |
| $q^h$ | Query embedding of the $h$-th attention head |
| $k^h$ | Key embedding of the $h$-th attention head |
| $v^h$ | Value embedding of the $h$-th attention head |

## H LIMITATION AND FUTURE WORK

Although CDIB outperforms the baseline models, it currently relies solely on ID features to model user attributes, and its ability to guide generating distributions is constrained by cold-start problems. In future work, we plan to investigate using side information or multi-modal data to model user attributes, which may help mitigate the cold start issues. Additionally, to maintain high computational efficiency, we employ a lightweight MLP model as the backbone for the denoising process. While the suitability of MLP for recommendation scenarios is not the focus of this work, it remains an important question. Therefore, we will explore which architectures are both lightweight and effective for recommendation scenarios, such as Mamba (Gu & Dao, 2024), in our future studies.

