# OpenReview forum: "Conditional Information Bottleneck Approach for Out-of-Distribution Sequential Recommendation"
_ICLR.cc/2025/Conference — Submitted to ICLR 2025_

### Official Review · Reviewer_VfYo · 2024-11-01

**Soundness:** 2
**Presentation:** 3
**Contribution:** 2
**Rating:** 5
**Confidence:** 3

**Summary:**

This research explores the out-of-distribution(OOD) problem faced in sequence recommendation within dynamic environments. Existing methods, such as Distributionally Robust Optimization (DRO) and random data augmentation, have been applied but often overlook the sensitivity of users to distribution changes and have shortcomings in data sparsity and information distortion.To address these issues, the authors propose a new framework called Conditional Distribution Information Bottleneck (CDIB), which aims to retain the minimal sufficient information related to users while generating diverse OOD distributions. The specific methods include utilizing a learnable masking mechanism to generate distributions in the latent space by masking stable elements and selecting elements sensitive to environmental factors. Also they try to minimize the mutual information between the original distribution and the generated distribution while maximizing the mutual information between the generated distribution and the target, ensuring the preservation of key user information.The method combines embedding layers, masking mechanisms, diffusion models, and Transformer recommendation systems in its architecture and optimization. Finally, the authors validate the effectiveness of their approach through various experiments.

**Strengths:**

Overall, the logic of the paper is quite clear, and the first half reads very cohesively. The proposed method is novel, and the detailed explanation of the model allows readers to easily understand the structure of the new framework presented in the paper.

**Weaknesses:**

1.The motivation for introducing the diffusion model is insufficient and feels forced. The mention of computational efficiency is misleading, as the computational efficiency of the diffusion model is actually quite low. The preliminary section lacks an introduction to the diffusion model.

2.The mathematical notation in the paper is confusing, with the meanings of superscripts and subscripts mixed up, such as $h_u^l$ and $h_u^0$ .Sometimes the superscript refers to the position of interaction, and at other times it refers to the timestep.

3.It is unclear how the stable and sensitive features are learned and what the ground truth is; the method does not explain this clearly.

4.The experimental section lacks a baseline for random data augmentation.

**Questions:**

1.The loss function for the learnable mask mechanism is somewhat strange, its purpose is not clearly defined. How does it manage to mask out the stable features?

2.The dataset does not use a distributionally shifted dataset but rather a conventional dataset. How can demonstrate that this method can address the problem of distribution shift?

---

### Official Review · Reviewer_zew3 · 2024-11-03

**Soundness:** 3
**Presentation:** 3
**Contribution:** 3
**Rating:** 6
**Confidence:** 4

**Summary:**

This paper designs the conditional distribution information bottleneck to alleviate the OOD problem in the sequential recommendation. Based on the principle, a diffusion-based framework optimizes the information bottleneck. Extensive experiments against various baselines demonstrate the effectiveness of the proposed method.

**Strengths:**

1. The motivation of the OOD problem in sequential recommendation makes sense.
2. The work effectively demonstrates both technical innovations and performance improvements over existing approaches, while providing a thorough and well-structured review of related literature.
3. The paper is well-written and easy to follow. The code is available.

**Weaknesses:**

1. The assumption in the proposed DIB seems unreasonable, and the authors do not give a reasonable explanation.
2. The proof of the proposition is disorganized and difficult to follow. For example, in Appendix A.1. The authors should reorganize it.
3. The expression of some theorems proposed in the appendix is not appropriate. For instance, the Theorem A.2 is an existing theorem, not the contribution of this work. Also, the authors do not give citations for this work.

**Questions:**

1. Could you explain the relevant variables in Figure 2?
2. Why do the stable elements $X_s$ and $Y_s$ of users and items follow the same distribution?
3. Please provide a corresponding explanation of the target Y in the main content.
4. Could you directly compare the generalization bound of CDIB with that of DRO methods, such as S-DRO[1] and DROS[2]?

Reference:
[1] Hongyi Wen, Xinyang Yi, Tiansheng Yao, Jiaxi Tang, Lichan Hong, and Ed H. Chi. Distributionally- robust recommendations for improving worst-case user experience. In The Web Conference (WWW), pp. 3606–3610, 2022.
[2] Zhengyi Yang, Xiangnan He, Jizhi Zhang, Jiancan Wu, Xin Xin, Jiawei Chen, and Xiang Wang. A generic learning framework for sequential recommendation with distribution shifts. In International Conference on Research and Development in Information Retrieval (SIGIR), pp. 331–340, 2023b.

---

### Official Review · Reviewer_AiPw · 2024-11-04

**Soundness:** 3
**Presentation:** 3
**Contribution:** 2
**Rating:** 5
**Confidence:** 3

**Summary:**

The paper proposes a novel approach, the Conditional Distribution Information Bottleneck (CDIB), for out-of-distribution (OOD) sequential recommendation systems, addressing dynamic environmental factors that cause performance degradation in traditional sequential recommendation (SR) models. CDIB, inspired by information bottleneck theory, aims to generate diverse distributions that preserve minimal sufficient information from the original distribution while incorporating user-specific distribution shifts. This method introduces a learnable mask-then-generate data augmentation strategy optimized with CDIB to enhance OOD robustness. Through experiments on four datasets, the authors demonstrate that CDIB outperforms baseline models in OOD contexts, showing improved adaptability to distribution shifts.

**Strengths:**

Sufficient Experiments:
The experimental part of this paper comprehensively and systematically demonstrates the effectiveness of the model. The author conducts experiments on multiple public data sets and compares the proposed model with multiple mainstream recommendation algorithms. In addition, the paper demonstrates the robustness of the model under distribution changes and different user preferences through rich experimental settings such as time distribution and user group experiments. The chart design is reasonable, which can clearly show the changing trend of model performance and its performance under different conditions, making the experimental results more convincing.

Detailed Theoretical Derivation:
The author fully mathematically deduces the theoretical basis and optimization objectives of the model, starting from the information bottleneck theory, gradually introduces the conditional information bottleneck (CDIB), and explains the model design logic through detailed formula description.

**Weaknesses:**

Component logic confusion:
The paper introduces multiple modules in the method section, including "Learnable Mask Mechanism", "Distribution Generator", etc., but their relevance to CDIB theory is not clear enough. For example, in Section 3.3, it is mentioned that "Learnable Mask first masks stable elements, and then the distribution generator enhances sensitive elements", but it does not explain why these steps are crucial to CDIB's conditional information bottleneck optimization goal. In addition, the introduction of CDIB and the specific logic between components lack intuitive explanations, making it difficult for readers to understand how these modules work together to achieve the goal of CDIB. The component design of the entire method section lacks a systematic discussion, making the overall structure of the model appear rather scattered.

Unclear theoretical motivation：
The article starts directly from DIB (distribution information bottleneck) in Section 3.1, but lacks an explanation of the shortcomings of the basic information bottleneck (IB) method in dealing with OOD (out-of-distribution data) problems, and does not fully explain the defects of the DIB method with distribution. Therefore, it is difficult for readers to understand why the basic IB cannot effectively deal with out-of-distribution problems, and why DIB still cannot meet the needs of the recommendation system after adding distribution information. The article does not elaborate on the motivation for introducing CDIB (Conditional Distribution Information Bottleneck), and fails to clearly explain how conditional information and distribution conditions solve the shortcomings of DIB in CDIB. The lack of such background explanation makes the proposal of CDIB seem unnatural and weakens the rationality of its theoretical innovation.

**Questions:**

1. I recommend that the authors provide an overview of how each component contributes to the goals of CDIB before diving into the technical details. Additionally, a diagram illustrating the relationships between components and their role in optimizing CDIB’s goals is recommended, which helps clarify the overall model structure.

2.I suggest that the authors add a brief subsection outlining the limitations of basic IB in dealing with OOD problems in recommender systems before introducing DIB. In addition, you can suggest that they clearly explain why DIB itself is not sufficient, thus naturally introducing CDIB. This will help readers better understand the progression of ideas and the necessity of CDIB.

---

### Official Review · Reviewer_qTt8 · 2024-11-05

**Soundness:** 3
**Presentation:** 3
**Contribution:** 3
**Rating:** 5
**Confidence:** 4

**Summary:**

To tackle the OOD problem in sequential recommendation scenarios, this paper proposes a novel conditional information bottlneneck approach(CDIB). Specifically, CDIB aims to preserve the minimal yet sufficient information when performing data augmentation. Based on the IB principle, the proposed CDIB can generate multiple distributions to enhance the robustness of the proposed approach.

**Strengths:**

1. The motivation is well illustrated, introducing IB principle to address OOD problem in SR is reasonable.
2. The proposed method is compelling and is backed by solid theoretical justification.
3. The availability of the code significantly contributes to the reproducibility of the results.

**Weaknesses:**

1.  For minimizing the mutual information I($D_{tr}$; $\tilde{D}$), why the authors chose the InfoNCE objective. InfoNCE is the lower-bound of mutual information, I think the authors should give the upper-bound for minimization.
2. It confused me what is the "user attributes" defined in the line 184. Based on the provided dataset description, I can't see any attribute information. If the "user attributes" in this paper denote user embeddings, I think this is not suitable, and easy to mislead the readers.
3. What are the benefits of the designed diffusion process? How it generate more ''sensitve elements''? As proposed in the above weakness, I really think the authors should use these nouns with caution. It seem the redundant of the diffusion based generation.
4. The overall optimization objective is too complex, the proposed approach is too complex, especially combining with the Transformer encoder and diffusion generation process.
5. Overall, IB principle is only optimization approach to remove the task-irrelevant information from the input sequence. How it performs on other backbones, can the authors provide more generality analysis of the propoased approach?
6. The used ML-100K is too small, why not conduct experiments on the larger ML datasets?

**Questions:**

See weaknesses

---

### Meta-Review · Area_Chair_hn8M · 2024-12-21

**Metareview:**

This paper focuses on the out-of-distribution problem in sequential recommendation scenarios and proposes the Conditional Distribution Information Bottleneck (CDIB) method based on Information Bottleneck theory. The motivation for the research was recognized by most reviewers, and the organization and presentation of the paper are relatively clear. However, three reviewers still considered the paper below the acceptance threshold, mainly due to the following key weaknesses: 1) All three reviewers raised significant concerns about the method's design and motivation, including Reviewer 1's skepticism about using InfoNCE; 2) The experimental setup is lacking, including the use of a small dataset and the absence of certain baseline types; 3) Some details and the proof process are unclear. Although the authors responded during the rebuttal phase, overall, the paper is not ready for publication by far.

**Additional Comments On Reviewer Discussion:**

During the rebuttal phase, some reviewers engaged in discussions with the authors, but the authors were unable to successfully convince the reviewers. Some reviewers did not respond, and after reviewing the relevant content, I think some of the weaknesses have still not been fully addressed.

---

### Decision · Program_Chairs · 2025-01-22

Reject